# NEURO-SYMBOLIC PROCEDURAL PLANNING WITH COMMONSENSE PROMPTING

**Yujie Lu**[1], **Weixi Feng**[1], **Wanrong Zhu**[1], **Wenda Xu**[1], **Xin Eric Wang**[2]
**Miguel Eckstein**[1], **William Yang Wang**[1]
[1]University of California, Santa Barbara, CA, USA
`{yujielu,weixifeng,wanrongzhu,wendaxu}@ucsb.edu`
`{migueleckstein,wangwilliamyang}@ucsb.edu`
[2]University of California, Santa Cruz, CA, USA
`xwang366@ucsc.edu`

## ABSTRACT

Procedural planning aims to implement complex high-level goals by decomposition into simpler low-level steps. Although procedural planning is a basic skill set for humans in daily life, it remains a challenge for large language models (LLMs) that lack a deep understanding of the cause-effect relations in procedures. Previous methods require manual exemplars to acquire procedural knowledge from LLMs in the zero-shot setting. However, such elicited pre-trained knowledge in LLMs induces spurious correlations between goals and steps, impairing the model's generalization to unseen tasks. In contrast, this paper proposes a neuro-symbolic procedural **PLAN**ner (PLAN) that elicits procedural knowledge from the LLMs with commonsense-infused prompting. To mitigate spurious goal-step correlations, we use symbolic program executors on the latent procedural representations to formalize prompts from external knowledge bases as a causal intervention toward the Structural Causal Model of procedural planning. Both automatic and human evaluations on WikiHow and RobotHow show the superiority of PLAN on procedural planning without further training or manual exemplars.[1]

## 1 INTRODUCTION

How to make a cup of coffee? As humans, we can easily specify a procedure to solve this task, using our innate ability of commonsense reasoning. However, can we endow machines with the same ability to construct a sequential plan? As depicted in Figure 1, procedural planning (Pearson, 1996; Zhang et al., 2020b; Huang et al., 2022) aims to decompose a high-level goal (Task: Watch TV) into a sequence of temporally extended steps (Procedural Plan: *Step* at all five time-steps).

We study procedural planning as the conditional text generation problem since it resembles real-world scenarios. Previous approaches (Huang et al., 2022; Ahn et al., 2022) require a small number of carefully written or held-out exemplars to acquire procedural knowledge. However, these manual exemplars evolved from task data are impossible to cover the ever-changing task setups and the flexible dependency relations among goals and steps. In fact, the biased data may cause the model to learn spurious correlations and hinder the model from generalizing well in zero-shot scenarios. Studies in cognitive science show that humans rely on chunking mechanisms (Gobet et al., 2001; Miller, 1956) which turn primitive stimuli into conceptual groups to solve novel and complex problems. Inspired by this, we hypothesize that generalizable procedural planning ability can be achieved by learning cause-effect relations among complex goals and simpler steps using external knowledge.

To reveal the cause-effect relations in procedural planning, we devise a Structural Causal Model (SCM) (Peters et al., 2017), a directed acyclic graph commonly used to describe the causal relationships within a system Pearl (2009). As depicted in Figure 2, the pre-trained knowledge ($D$) (e.g., TV and living room is highly correlated) in LLMs confounds ($D$ influences $T$, $S_{i-1}$ and $S_i$, resulting in spurious correlations) the system to make biased decisions toward an unreasonable step (e.g., Find

---

[1]Source code and datasets are publicly available at https://sites.google.com/view/iclr-clap

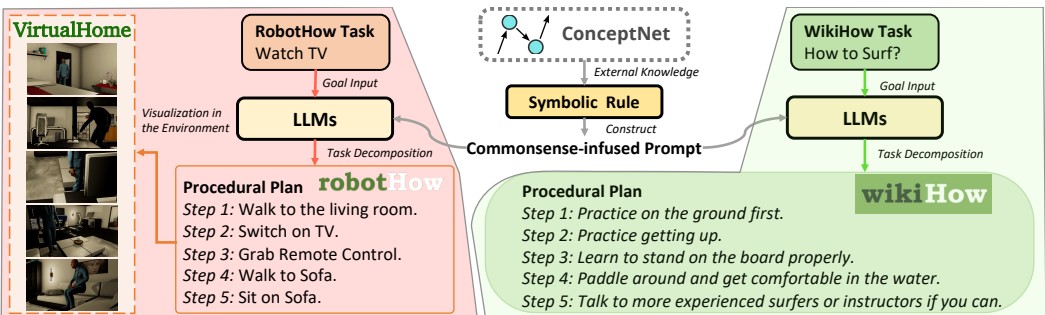

Figure 1: Two independant procedural planning task examples from RobotHow and WikiHow. PLAN construct commonsense-infused prompt from external knowledge (e.g., ConceptNet) to elicit procedural planning ability of the Large Language Models (LLMs) without training or exemplars.

Television). Thus, we adopt front-door adjustment (definition in Appendix A.3), which utilizes a mediator ($P_i$) that blocks all directed paths from the cause ($T$ or $S_{i-1}$) to the effect ($S_i$). In this way, $T$ (or $S_{i-1}$) affects $S_i$ by flowing through indirect paths: $T$ (or $S_{i-1}$) affects $P_i$ and $P_i$ affects $S_i$. And we can identify the causal effects among goals and steps by investigating the indirect effect (Equation 3), which is computed by multiplying the effect of $T$ (or $S_{i-1}$) on $P_{i-1}$ (Equation 1) with the effect of $P_i$ on $S_i$ (Equation 2). With the above front-door adjustment, we can mitigate the spurious correlations (e.g., between "television" and "living room") and thus make reasonable decisions on steps (e.g., Find book). Please refer to A.1 for causal preliminaries (including explanation for SCM, confounder, mediator, spurious correlations), and A.3 for the front-door adjustment definition.

Guided by the above causal analysis of procedural planning, we need to construct the mediator $P_i$ and then intervene on task $T$ and prompt $P_i$, which is required to compute the conditional probability in Equation3. As depicted in Figure 3, we seek to automatically construct commonsense-infused prompts as the mediator $P_i$ by concatenating the task, previous steps with commonsense knowledge extracted from external resources (e.g., ConceptNet (Speer et al., 2017)). First, we modify the goal input by sampling a task-relevant knowledge subgraph (`Stage1` in Section 3.1) to implement interventions on $T$. Then, we modify the prompt by adapting the edge weight to implement interventions on $P_i$ (Edge-Wise Adoption of `Stage2` in Section 3.1). However, directly incorporating knowledge of graph structure into LLMs leads to the loss of the logical order in eliciting procedural knowledge from LLMs. Thus, we apply symbolic executors (Mao et al., 2019; Yi et al., 2018) that execute the sequential mapping program on latent knowledge representations (e.g., the subevent of). In this way, we transit graph structure knowledge into natural language that preserves procedural structure, such as the sequential order of two low-level steps (Symbolic Structuring of `Stage2` in Section 3.1). The procedural prompt $P_G$ (e.g, "please get the remote control") is further translated into admissible one $\hat{P_G}$ (e.g., "grab remote control") from available steps in a certain domain (RobotHow or WikiHow in our case). Finally, we utilize the commonsense-infused prompt $\hat{P_G}$ to control the generation of procedural plans in LLMs in a zero-shot setting (Section 3.2).

We conducted experiments on RobotHow (Puig et al., 2018) and WikiHow (Koupaee & Wang, 2018) under original and counterfactual situations. Our major contributions can be summarized as:

- We develop the first causal framework for procedural planning by 1) defining a temporally extended Structural Causal Model and 2) resolving spurious correlation between high-level goals and low-level steps via front-door adjustment with a prompt-based mediator.
- We propose a neuro-symbolic approach to construct commonsense-infused prompts for LLMs to tackle the procedural planning task without manual exemplars or further training.
- Extensive evaluations show the superiority of PLAN in terms of reasoning about the cause-effect relations among goals and steps and achieving promising planning ability.

## 2 EXTERNAL KNOWLEDGE MATTERS IN PROCEDURAL PLANNING

As depicted in Figure 1, procedural planning requires generating the Plan (e.g., *Step 1:* Walk to the living room.) conditioned on the Task (e.g., Watch TV). We first describe the problem definition

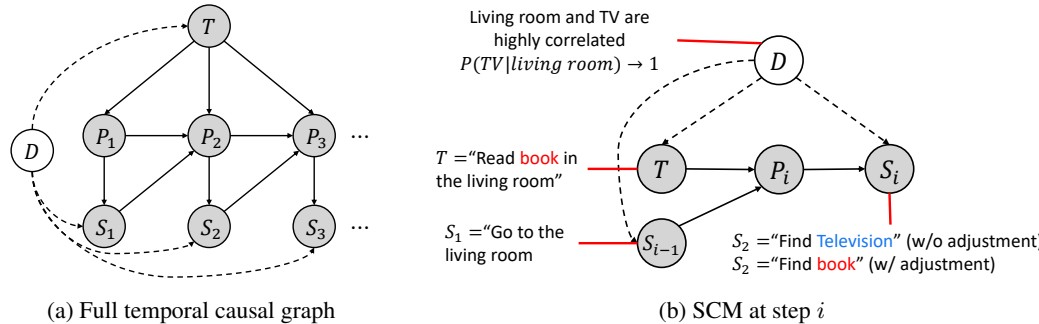

(a) Full temporal causal graph          (b) SCM at step $i$

Figure 2: **Structural Causal Model (SCM) for Procedural Planning.** (a) The full temporal causal graph. $T$ denotes the task query, and $S_i$ is the sub-goal step at timestep $i$. $D$ is the unobservable confounding variable introduced by the LLMs. $P_i$ denotes the mediating variables we construct to mitigate the spurious correlation. (b) The SCM at timestep $i$. Without causal intervention, the model produces a sub-goal step "find television" due to the spurious correlation between "television" and "living room" caused by the confounding variable $D$. With our causal intervention, the constructed mediating variable $P_i$ (Section 3.1) can block the backdoor paths for $T \rightarrow S_i$ and $S_{i-1} \rightarrow S_i$ (opened by $D$) and generate the causal sub-goal "find book" precisely (Section 3.2).

and then show why external knowledge matters in procedural planning through the lens of causality. Finally, we show how we elicit procedural ability from the Large Language Models (LLMs).

## 2.1 PROBLEM DEFINITION

Given the high-level task $T$ (e.g. watch television in the living room) sampled from a task domain $M_T$ (e.g. RobotHow), a procedural planner aims to decompose it into lower-level temporally extended steps $S_T = \{S_1, ..., S_i | S_i \in \bar{S}\}$. There exists certain admissible plans $\bar{S}$, which is a fixed set constrained by the task domain $M_T$ (e.g., the affordance of the interacted objects). The plan $S_i$ at timestep $i$ is generated as $\pi(S_i | T, S_{0:i-1})$.

## 2.2 A CAUSAL LOOK AT PROCEDURE PLANNING WITH LLMS

We seek to empower the LLMs with the ability to reason cause-effect relations in procedural planning. Thus, we devise a causal framework by first defining a Structural Causal Model (SCM) of procedural planning in Figure 2. The SCM describes the temporal dynamics and procedural cause-effect relationship. Our causal assumption in SCM indicates that there is a backdoor path from task to step, which must be blocked with front-door adjustment. Therefore, we model the input prompt as a mediator which is created from external knowledge. More specifically, we define our Full Temporal Causal Graph as in Figure 2a, which is an unrolled Structural Causal Model (SCM) for sequential decision-making. Our goal is to identify the causal relations between the attended task $T$ and plan procedures $S_T = \{S_1, S_2, ...\}$ from LLMs. Initially, there are direct paths $T \rightarrow S_i$ and $S_k \rightarrow S_i, k < i$ because $S_i$ relies on the LLM attended task entities and previous accomplished steps. $D$ is an unobserved confounder from learned knowledge during pre-training. $D$ builds a backdoor path between $T$ and $S_i$ and misguides the LLMs to attend to false entities to generate the next step (see Fig. 2b). Note that $D$ is unobservable as we directly adopt the LLM without knowing the pre-training data. To mitigate the spurious correlation, we then introduce a mediator $P_i$ for each $S_i$ as shown in Figure 2a. To achieve our front-door adjustment, we inject external knowledge into LLMs with a neuro-symbolic approach by adopting three stages described in Section 3.1.

## 3 OUR APPROACH

Although LLMs have strong general language intelligence, they still perform poorly in reasoning the cause-effect relations in procedural plans due to a lack of daily life experience. We propose to elicit the unbiased procedural planning knowledge from the LLMs using the created commonsense-infused Prompt $P$ as $\pi(S_i | T, S_{0:i-1}, P)$. Figure 3 and Algorithm 1 depict how PLAN tackles the procedural

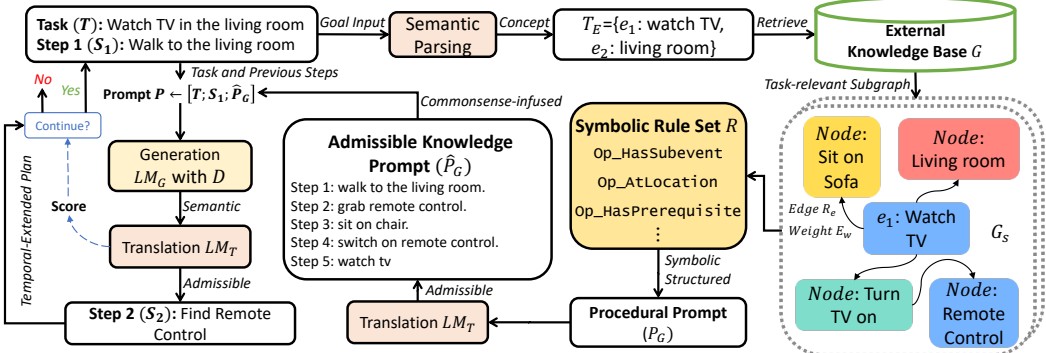

Figure 3: **The Overview of Procedural Planning.** Our five-stage pipeline includes: 1) semantically parsing the task $T$ into concept set $T_E$ to retrieve subgraph $G_s$ from the external knowledge base $G$. 2) formalize procedural prompt $P_G$ and then translate into the admissible one $\hat{P}_G$. 3) aggregate task, previous steps and $P_G$ as final commonsense-infused prompt $P$. (Section 3.1) 4) and 5) generating and translating time-extended procedural plans until triggering the termination condition. (Section 3.2)

planning in a five-stage manner. We illustrate the commonsense-infused prompt construction (the first three stages) in Section 3.1 and planning with LLMs (the last stage) in Section 3.2.

## 3.1 COMMONSENSE-INFUSED PROMPT CONSTRUCTION

**Overview** Inspired by the causal analysis in Section 2.2, we propose to construct commonsense-infused Prompt $P$ that helps reveal the cause-effect relations among the goals and steps during procedural planning within 3 stages: 1) Stage1 sample a subgraph $G_s$ from the external knowledge base $G$ by extracting task($T$)-relevant nodes. 2) Stage2 adapt the edge weight $E_w$ in $G_s$ and apply symbolic structuring to get the admissible knowledge prompt $\hat{P}_G$. 3) Stage3 acquire the temporal order by temporally aggregated the prompt $P_i$ with previous steps $S_{0:i-1}$.

**Stage1 : Task-Relevant Knowledge Subgraph Sampling** First, we investigate the causal effect $T \to P_i$ and $S_{i-1} \to P_i$ (Figure 2). $S_i$ is a collider that blocks the association between $D$ and $P_i$ in the path $T \leftarrow D \to S_i \leftarrow P_i$. Let $\pi_i$ denote $\pi(\cdot|P_{i-1})$ that represent the probability density function conditioned on $P_{i-1}$. Since there is no backdoor path for $T \to P_i$ and similarly for $S_{i-1} \to P_i$, we simply have the conditional probability after applying *do*-operators:

$$\pi_i(P_i = p|do(T)) = \pi_i(P_i = p|T), \quad \pi_i(P_i = p|do(S_{i-1})) = \pi_i(P_i = p|S_{i-1}) \tag{1}$$

We achieve the *do*-operation in a prompting way by modifying the goal input so that the model attends to the task-relevant entities. To implement, we use NLTK to tokenize and `pos_tag` the task text $T$. Then we use the noun (e.g. television), noun phrases (e.g. remote control), and verb phrases (e.g. watch television) as the concept node. In this way, the task name $T$ is Semantically Parsed into the Concept Set $T_E$. Each concept $e \in T_E$ is used as a query for sampling the $H$-hop task-relevant subgraph $G_s \subseteq \mathcal{N}_e \times \mathcal{R}_s \times \mathcal{N}_e$ from the external knowledge base $G \subseteq \mathcal{N} \times \mathcal{R} \times \mathcal{N}$, where $\mathcal{N}$ and $\mathcal{R}$ represent the number of concept nodes and commonsense relations respectively. When extracting $G_s$, we keep the triplets with relation type in the household domain (e.g., `AtLocation`, `UsedFor`) and filter out ones in the linguistic domain (e.g., `DistinctFrom`, `DerivedFrom`) for the procedural planning task. $\mathcal{N}_e$ is maintained in a set of top-$k$ task-relevant nodes using the weight of each $R_e$, which is updated with edge-wise adaption in Stage2.

**Stage2 : Edge-Wise Adaption and Symbolic Structuring** Second, we need to find the causal effect for $P_i \to S_i$. Since the path $P_i \leftarrow T \leftarrow D \to S_i$ contains a backdoor from $P_i$ to $S_i$, we cannot rely on the conditional probability. Instead, we intervene on $P_i$ using *do*-operator to cut off $D \to T$:

$$\pi_i(S_i|do(P_i = p)) = \sum_{t,s} \pi_i(S_i|p, T = t, S_{i-1} = s)\pi_i(T = t, S_{i-1} = s)$$

$$= \sum_{t,s} \pi_i(S_i|p, T = t, S_{i-1} = s)\pi_i(S_{i-1} = s|T = t)\pi_i(T = t) \tag{2}$$

The retrieved concept-centered graph has multiple edges representing various relationships with other actions/entities. Therefore, the summation over intervened $T$ can be achieved by incorporating these edges into the prompt. For instance, "living room" can be "walked to" and "used for reading" while "book" can locate in "living room" and "bedroom". Similarly, we extrapolate over the edges for $i - 1$ hops to aggregate the intervened $S_i$, i.e. $P(S_{i-1} = s | T = t)$. Directly ranking the retrieved nodes $N_e$ with the annotated weight ($E_w$) in the external knowledge base will result in a spurious correlation. Because such retrieved local subgraphs tend to capture the task-invariant concept nodes as the causal factors. To mitigate this, we propose to adapt the weight of each triplet (**Edge-wise Adaption**). The adapted weight is the addition of the original edge weight and the cosine similarity between the tail node embedding $\boldsymbol{n}_{E_{tail}}$ of the edge $R_e$ and the task embedding $\boldsymbol{v}_{task}$ as: $\hat{E_w} \leftarrow E_w + cosine(\boldsymbol{n}_{E_{tail}}, \boldsymbol{v}_{task})$. The embeddings are projected from the node text and task name using the sentence-transformer (Reimers & Gurevych, 2019). The nodes $N_e$ are finally retrieved by ranking the adapted weight $\hat{E_w}$. To better track the utilized external knowledge during inference, we construct the task-dependent commonsense prompt with a Symbolic Executor (**Symbolic Structuring**) guided by the relation type of each triplet in $G_s$ with the adapted edge weight beyond threshold $\theta_e$. Specifically, the Symbolic Executor acquires the neural information of each natural language node and executes the sequential mapping program by sampling the operation $Op$ from the Symbolic Rule Set $R$ according to the edge relation type. The Symbolic Rule Set $R$ is obtained by mapping the description of the relations (e.g., $AtLocation$ represent 'A is a typical location for B, or A is the inherent location of B. Some instances of this would be considered meronyms in WordNet.') in the external knowledge graph (e.g., ConceptNet) to symbolic operations (e.g., $Op\_AtLocation$). For instance, the $AtLocation$ edge samples the operation `Op_AtLocation` from $R$, which takes the commonsense relation of the triplet from $G_s$ as the parameters to query the procedural concept output given the natural language meaning of the linked nodes (e.g., go to the location of `Start_Node_Of`$(r_e)$ in this case). Similarly, `Op_UsedFor` may refer to "go to find `End_Node_Of`$(r_e)$ and use it for `Start_Node_Of`$(r_e)$". And operators `Op_HasSubevent` and `Op_HasPrerequisite` will recursively navigate the subgraph $G_s$. After navigating the subgraph, we linearize the transformed triplets as the Procedural Prompt $P_G$, which is then translated to Admissible Knowledge Prompt $\hat{P}_G$ by the Translation Language Model $LM_T$.

**Stage3: Temporally-Extended Aggregation** To acquire temporal order in the procedure, we obtain the Prompt $P$ at timestep $i$ with the aggregation of task $T$, history steps $S_{0:i-1}$ and current external knowledge $\hat{P}_G$. The underlying causal mechanism is a combination of Eq. 1 and Eq. 2:

$$\begin{aligned} \pi_i(S_i | do(T), do(S_{i-1})) &= \sum_p \pi_i(S_i | do(P_i = p)) \pi_i(p | do(T), do(S_{i-1})) \\ &= \sum_p \pi_i(p|T) \sum_{t,s} \pi_i(S_i | p, T = t, S_{i-1} = s) \pi_i(T = t, S_{i-1} = s) \end{aligned} \tag{3}$$

The adjustment and marginalization in Eq. 3 is achieved in the input space by forming the Procedural Prompt $P_G$ that allows the LLM to attend on the causal entities instead of the highly correlated ones for the next step generation. The LLM can reason over the most relevant edges to link the concepts with the task entities as a context. The prompts from knowledge bases are independent of the pre-training data distribution so that $P_i$ is independent of $D$ and satisfies the front-door criterion. Please refer to Appendix A.3 and Figure 4 for the simplification of our structural causal model.

## 3.2 PROCEDURAL PLANNING WITH LARGE LANGUAGE MODELS

**Stage4: Semantic Generation** The external knowledge is further concatenated with the goal input ($T$) as the initial prompt. Given the prompt, the language model Generation $LM_G \in \{P_{AR}, P_{AE}\}$ (e.g., GPT3, BART) generates the next sentence, and the most confident prediction is then appended to previous prompts. The Termination Condition is either reaching the max step $t$ or the matching score is below threshold $\theta$. The joint probabilities of auto-regressive ($P_{AR}$) and auto-encoder ($P_{AE}$) model is factorized as:

$$\pi_{AR}(x) = \prod_{i=1}^n p(s_n | \hat{P}_G, s_{1:n-1}, T), \quad \pi_{AE}(x) = \prod_{i=1}^n p(s_n | \hat{P}_G, \{s_{1:n-1}, [MASK]\}, T) \tag{4}$$

where $\hat{P}_G$ represent the commonsense knowledge and $T$ represent the task name.

---

**Algorithm 1** Neuro-Symbolic Procedural Planning using Commonsense-Infused Prompting

---

**Require:**

   Task Sample $T$, Admissible Step Set $S$, External Knowledge Graph $G$;
   Language Model for Generation $LM_G$ and Translation $LM_T$, Symbolic Rule Set $R$;

**Ensure:**

 1: `[Stage1]` Semantically parse $T$ into entity set $T_E$;
 2: Maintain top-$k$ task-relevant nodes $\mathcal{N}_e$ in $T_E$;
 3: Retrieve subgraph $G_s \subseteq \mathcal{N}_e \times \mathcal{R}_s \times \mathcal{N}_e$ from $G \subseteq \mathcal{N} \times \mathcal{R} \times \mathcal{N}$ for each $e \in T_E$;
 4: `[Stage2]` Edge-wise adaption as $\hat{E}_w \leftarrow E_w + cosine(n_{E_{tail}}, v_{task})$ and re-rank $\mathcal{N}_e$ in $T_E$;
 5: Map the description text of the relations $\mathcal{R}_s$ in $G_s$ as Symbolic Rule Set $R$;
 6: Construct procedural prompt $P_G$ by verbalizing the re-weighted $G_s$ using $R$;
 7: Translate $P_G$ in Admissible Knowledge Prompt $\hat{P}_G = LM_T(P_G)$;
   Temporally-extended zero-shot inference for Procedural Plan $S_T = \{S_1, ..., S_i\}$:
 8: **for** each timestep $i$ **do**
 9:    `[Stage3]` Aggregate Prompt $P_i \leftarrow [T; S_{0:i-1}; \hat{P}_G]$;
10:    `[Stage4]` and `[Stage5]` $S_i = LM_T(LM_G(P_i))$;
11:    Update Procedural Plan $S_T \leftarrow S_i$;
12: **end for**

---

**Stage5: Admissible Step Translation** To ensure that the generated procedural plans are grounded to the environment, we should avoid producing the steps that are inadmissible (e.g. Toast the table). In other words, the generated steps should be fully constrained to the admissible composite of action and object in a certain task domain. Thus previous works (Huang et al., 2022; Ahn et al., 2022) have explored using the model (which is $LM_T$ in our case) to score a step selected from a fixed set of available options, instead of directly sampling from the output distributions of the language model (which is $LM_G$ in our case). Specifically, we match the generated step by $LM_G$ to the most similar admissible step in the embedding space encoded by the Translation Language Model $LM_T$. Following (Huang et al., 2022), we utilize a Sentence-Transformer (Reimers & Gurevych, 2019) to calculate the cosine similarity as $\pi(s_i|x) = LM_T(LM_G(x))$, which translates $LM_G(x)$ into the admissible step $s_i \in \bar{S}$ that is the closest in the embedding space measured by the cosine similarity.

### 3.3 Counterfactual Procedural Data Construction

To investigate the counterfactual reasoning ability, we design three families of intervention methods: 1) **Initial Configuration**: intervene in the initial configuration, such as the location for implementing the task. 2) **Intermediate Step**, randomly select one step from the ground truth program as an additional constraint of implementing the task and append it to the task name for generating the procedural plan. 3) **Final Goal**, intervene the task goal as the composite of another randomly sampled task. Table 5 in the Appendix summarizes the category and description. The counterfactual dataset construction details and post-intervention examples are provided in Appendix B.2.

## 4 Experiments

### 4.1 Procedural Planning Setup

**Datasets** We conduct zero-shot experiments on two datasets with procedural information, WikiHow (collected following (Koupaee & Wang, 2018)) and RobotHow (Puig et al., 2018) without training. **WikiHow** is a large-scale text summarization dataset that is constructed from a human-written knowledge base, involving procedural tasks that spans various topics. We utilize "how to" title as the task names and the summarized headlines as the steps. **RobotHow** is a large knowledge base of common household tasks collected in the VirtualHome (Puig et al., 2018) simulator. The dataset contains the programs with high-level task names and low-level steps. $M_T$ is composed of 292 and 2000 distinct tasks from RobotHow and WikiHow respectively. Human evaluations use randomly sampled 50 task examples for each dataset. Automatic evaluations use 150 and 1000 task examples randomly sampled from RobotHow and WikiHow respectively. Please refer to Appendix B.1 and Appendix B.2 for dataset details.

| Dataset | Model$_{base}$ | Original-Coverage | | | Original-Order | | | Counterfactual-Coverage | | | Counterfactual-Order | | |
|---|---|---|---|---|---|---|---|---|---|---|---|---|---|
| | | Win(↑) | Tie | Lose(↓) | Win(↑) | Tie | Lose(↓) | Win(↑) | Tie | Lose(↓) | Win(↑) | Tie | Lose(↓) |
| **RobotHow** | BART (Lewis et al., 2020) | **46.67** | 31.33 | 22.00 | **50.00** | 22.67 | 27.33 | **42.00** | 22.67 | 35.33 | **50.00** | 18.67 | 31.33 |
| | GPT2 (Radford et al.) | **42.67** | 22.00 | 35.33 | **44.00** | 18.67 | 37.33 | **56.67** | 11.33 | 32.00 | **45.33** | 16.00 | 38.67 |
| | GPT3 (Brown et al., 2020) | **50.00** | 23.33 | 26.67 | **53.33** | 23.33 | 23.33 | **54.67** | 16.67 | 28.67 | **56.00** | 15.33 | 28.67 |
| **WikiHow** | BART (Lewis et al., 2020) | **56.67** | 12.67 | 30.67 | **69.33** | 10.00 | 20.67 | **50.00** | 26.67 | 23.33 | **46.00** | 21.33 | 32.67 |
| | GPT2 (Radford et al.) | **48.00** | 16.00 | 36.00 | **49.33** | 11.33 | 39.33 | **46.67** | 16.67 | 36.67 | **44.67** | 19.33 | 36.00 |
| | GPT3 (Brown et al., 2020) | **75.17** | 10.74 | 14.09 | **72.67** | 8.67 | 18.67 | **44.00** | 22.67 | 33.33 | **48.67** | 25.33 | 26.00 |

Table 1: Percentages of procedural planning results of PLAN that are better than, tied with, or worse than Planner (Huang et al., 2022), in coverage and order metrics under the original and counterfactual setting.

| Architecture | Model | RobotHow | | | | WikiHow | | | |
|---|---|---|---|---|---|---|---|---|---|
| | | Original | | Counterfactual | | Original | | Counterfactual | |
| | | Coverage | Order | Coverage | Order | Coverage | Order | Coverage | Order |
| **BART (Lewis et al., 2020)** | Chain (Wei et al., 2022) | 2.99 | 2.80 | 2.71 | 2.76 | 2.88 | 3.42 | 3.34 | 2.97 |
| | LLMaP (Huang et al., 2022) | 3.06 | 2.84 | 2.96 | 2.82 | 2.78 | 3.35 | 3.46 | 3.02 |
| | PLAN (Ours) | **3.16** | **3.10** | **3.07** | **2.98** | **3.05** | **3.47** | **3.62** | **3.18** |
| **GPT2 (Radford et al.)** | Chain (Wei et al., 2022) | 2.43 | 2.28 | 3.12 | 2.88 | 2.97 | 3.44 | 3.60 | 3.01 |
| | LLMaP (Huang et al., 2022) | 3.09 | 2.94 | 2.93 | **2.90** | 3.20 | 3.53 | 3.63 | 3.24 |
| | PLAN (Ours) | **3.12** | **2.99** | **3.43** | 2.88 | **3.67** | **3.69** | **3.81** | **3.31** |
| **GPT3 (Brown et al., 2020)** | Chain (Wei et al., 2022) | 3.26 | 3.18 | 3.45 | 3.58 | 3.29 | 3.46 | 3.70 | 3.71 |
| | LLMaP (Huang et al., 2022) | 3.50 | 3.56 | 3.56 | 3.53 | 3.21 | 3.27 | 3.77 | 3.71 |
| | PLAN (Ours) | **3.72** | **3.70** | **3.67** | 3.56 | **3.72** | **3.82** | **3.85** | **3.75** |

Table 2: Averaged 5-point Likert scale human evaluations on "coverage" and "order" aspects.

**Baselines** We compare our approach with three vanilla generative pre-trained language models (BART, GPT2, and GPT3) and two powerful generation baselines (Zero-shot Planner (Huang et al., 2022) noted as "LLMaP" and Chain of Thought (Wei et al., 2022) noted as "Chain"). More method and configuration details of the models can be found in Appendix B.3 and Appendix B.4.

**Metrics** We ask human annotators on the Amazon Mechanical Turk platform to rate model performance on two aspects: 1) `Coverage`: depicts which set of steps can better complete the target task (captures semantic completeness). 2) `Order`: depicts which sequence covers more steps that are necessary to complete the target task (captures sequential order correctness). In addition, we use Sentence-BLEU (S-BLEU) (Papineni et al., 2002), BERTScore (Zhang* et al., 2020), ROUGE-1 (Lin, 2004) and Word Mover's Distance (WMD) (Kusner et al., 2015) as automatic evaluation metrics. These metrics are used to compute the semantic scores between the annotated programs and the predictions. Details of the crowdsourcing human evaluation can be found in Appendix C.1.

## 4.2 HUMAN EVALUATION RESULTS WITH COVERAGE AND ORDER METRIC

Each example is rated by 3 crowdsourcing annotators. For the **Win-Lose Comparison**, we ask the human rater to choose between ours and the baseline LLMaP (Huang et al., 2022). Averaged results reported in Table 1 show that our PLAN is more frequently rated as better for both coverage and order metrics, outperforming baselines over the winning ratio by 21% in coverage and 26% in order, across two datasets. We report the average results of **Human Ratings** with 5-point Likert scale in Table 2. The consistent performance boost of PLAN indicates the superiority of injecting external commonsense knowledge into the procedural planning task. The performance drop of LLMaP and Chain in the counterfactual setting indicates the vulnerability of fixed holdout knowledge and the pre-defined manual exemplars in causal procedural planning. Please refer to Appendix C.1 for the crowdsourcing human evaluation interface details. Table 3 shows two examples for **Qualitative Comparison**. More examples can be found in Appendix D.

## 4.3 AUTOMATICALLY MEASURING THE PROCEDURAL PLANNING

**Main Results** Table 4 summarizes The automatic evaluation results. PLAN achieves the best results regardless of the architecture of the language model architecture, either autoregressive or autoencoder based. The performance gain of "LLMaP" over "Chain" may probably be due to direct exposure to the holdout task from the dataset. While the "Chain" baseline still outperforms the vanilla baseline that only takes the high-level task name as the prompt. Note that the annotated program is not the

| Dataset | Task | Model | Steps |
|---|---|---|---|
| **RobotHow** | Toast Bread | Chain | Step 1: Walk to kitchen. Step 2: Put bowl on oven. Step 3: Switch off oven. Step 4: Open oven. Step 5: Grab bowl. Step 6: Put bowl in oven. |
| | | LLMaP | Step 1: Put bread in oven. Step 2: Put food bread in oven. Step 3: Put food bread on table. |
| | | PLAN | Step 1: Walk to kitchen counter. Step 2: Pull bread slice. Step 3: Put bread slice on toaster. Step 4: Switch on toaster. Step 5: Look at toaster. Step 6: Turn to toaster. |
| **WikiHow** | How to Take Care of a Pet | Chain | Step 1: Take care of your pets. Step 2: Scroll to the end of the text file. |
| | | LLMaP | Step 1: Make sure your pet always has enough food and water. Step 2: Be sure to feed your dog at least three times a day. Step 3: Feed your pet high-quality "complete and balanced" dog food. Step 4: Give your dog fruits and vegetables as snacks. Step 5: Eat a variety of food. Step 6: Feed your puppy healthy food. |
| | | PLAN | Step 1: Research the type of pet, so you can make sure the pet fits into your environment and lifestyle. Step 2: Choose your breed carefully, especially if you have children. Step 3: Exercise your pet if it's the kind of pet that needs it, such as a dog. Step 4: Watch your pet's diet carefully. Step 5: Monitor your pet for illness and injury. Step 6: Make time for your pet. |

Table 3: Showcases of procedural steps predicted by different models with GPT2 as the base LLM.

| Model | RobotHow | | | | WikiHow | | | |
|---|---|---|---|---|---|---|---|---|
| | S-BLEU | WMD | BERT-f1 | ROUGE-f1 | S-BLEU | WMD | BERT-f1 | ROUGE-f1 |
| BART Lewis et al. (2020) | 0.069 | 0.923 | 0.870 | 0.442 | 0.083 | 0.937 | 0.836 | 0.379 |
| BART + Chain (Wei et al., 2022) | 0.079 | 0.913 | 0.862 | 0.448 | 0.095 | 0.939 | 0.782 | 0.377 |
| BART + LLMaP (Huang et al., 2022) | 0.094 | 0.940 | 0.870 | 0.467 | 0.131 | 0.950 | 0.816 | 0.371 |
| BART + PLAN (Ours) | **0.110** | **0.951** | **0.890** | **0.528** | **0.142** | **0.958** | **0.833** | **0.400** |
| $w/o$ Adaption | 0.104 | 0.929 | 0.886 | 0.492 | 0.132 | 0.952 | 0.824 | 0.398 |
| $w/o$ Symbolic | 0.062 | 0.858 | 0.835 | 0.392 | 0.087 | 0.939 | 0.828 | 0.386 |
| GPT2 (Radford et al.) | 0.056 | 0.891 | 0.846 | 0.356 | 0.051 | 0.925 | 0.826 | 0.345 |
| GPT2 + Chain (Wei et al., 2022) | 0.079 | 0.906 | 0.861 | 0.405 | 0.124 | 0.937 | 0.817 | 0.352 |
| GPT2 + LLMaP (Huang et al., 2022) | 0.115 | 0.931 | 0.885 | 0.481 | 0.115 | 0.957 | 0.833 | 0.363 |
| GPT2 + PLAN (Ours) | **0.148** | **0.945** | **0.898** | **0.547** | **0.133** | **0.971** | **0.835** | **0.373** |
| $w/o$ Adaption | 0.142 | 0.944 | 0.896 | 0.542 | 0.123 | 0.965 | 0.830 | 0.360 |
| $w/o$ Symbolic | 0.143 | 0.942 | 0.895 | 0.538 | 0.121 | 0.967 | 0.829 | 0.357 |
| GPT3 (Brown et al., 2020) | 0.072 | 0.882 | 0.855 | 0.416 | 0.077 | 0.936 | 0.832 | 0.366 |
| GPT3 + Chain (Wei et al., 2022) | 0.089 | 0.905 | 0.860 | 0.471 | 0.094 | 0.943 | 0.839 | 0.393 |
| GPT3 + LLMaP (Huang et al., 2022) | 0.123 | 0.931 | 0.894 | 0.539 | 0.116 | 0.946 | 0.842 | 0.401 |
| GPT3 + PLAN (Ours) | **0.155** | **0.939** | **0.902** | **0.561** | **0.155** | **0.961** | **0.849** | **0.433** |
| $w/o$ Adaption | 0.139 | 0.923 | 0.887 | 0.517 | 0.144 | 0.955 | 0.830 | 0.420 |
| $w/o$ Symbolic | 0.135 | 0.933 | 0.898 | 0.536 | 0.140 | 0.959 | 0.843 | 0.414 |

Table 4: Automatic evaluation results on the Original RobotHow and WikiHow. Metrics are computed between the annotated programs and the predictions.

only solution, thus these automatic metrics provide limited absolute performance information. Details for the correlation between automatic metrics and human evaluation can be found in Section 4.5.

**Effects of Edge-wise Adaption and Symbolic Program Execution** The variant "$w/o$ Adaption" maintains the top-$k$ task-specific nodes ranked by the annotated weight $E_W$ in the external knowledge base $G$ without adaption. The variant "$w/o$ Symbolic" directly takes the extracted concept nodes from external knowledge base as prompt. The performance drop of these two variants in Table 4 with significance test in Appendix C.2 demonstrate the importance of adaption and symbolic modules.

**Effects of the Large Language Model Architecture** We use GPT2 and GPT3 as autoregressive architecture and BART (Lewis et al., 2020) as autoencoder architecture. The autoregressive architecture achieves better results than the autoencoder one. Since the pre-training objective of autoregressive-based GPT is to predict the next token given the previous input tokens. We assume the performance gain of GPT is due to a smaller gap between the objective of pre-training and procedural planning.

**Level of Complexity** We show report results that use the test set which is separated into several buckets according to the number of steps in the procedural planning task. The step number reflects the difficulty of the task. In Table 7 and Table 8 in Appendix C.2, we show that the averaged performance gain of PLAN over the baselines are consistent or more significant in more complicated procedural planning settings. This indicates the superiority of PLAN in solving long-horizon tasks.

## 4.4 RESULTS ON COUNTERFACTUAL TASK SAMPLES

We apply *Initial Configuration*, *Intermediate Step*, *Final Goal* interventions on RobotHow and *Intermediate Step* on WikiHow. Human evaluations under counterfactual setting are summarized in Table 1 and Table 2. PLAN consistently outperforms baselines by a large margin and experiences

a much smaller performance drop compared with the powerful baselines when switching to the counterfactual setting. We assume it's due to the biased knowledge of the holdout examples and manual exemplars utilized in the baselines, which are vulnerable to counterfactual samples. Automatic evaluations on counterfactual RobotHow are summarized in Table 13 in Appendix C.2. Aligned with human evaluations, PLAN achieves the best performance. The overall poor performance in *Final Goal* category indicates the challenge for long-horizon and composite procedural planning. While the overall better performance in *Intermediate Step* category benefits from the intermediate guidance.

## 4.5 Correlation between Automatic and Human Evaluation

We evaluate segment-level **Pearson Correlation** between human and automatic metrics. We observe that BERTScore has a moderate correlation to the human coverage score and WMD has a moderate correlation to the human order score, with 23.3% and 32.3% respectively. Similar to the prior findings (Xu et al., 2021), n-gram-based metrics (Sentence-BLEU and ROUGE) have a relatively weaker correlation to the human coverage score, with a Pearson correlation of 16.4% and 21.1%. Overall, our automatic and human evaluation scores are consistent with the main claim of this paper. However, human evaluation is still irreplaceable for procedural planning at the current stage.

## 5 Related Work

**Procedural Planning** Learning to generate procedural plan (Zhang et al., 2020a; Lyu et al., 2021; Zhang et al., 2020b; Chang et al., 2020; Wu et al., 2022; Huang et al., 2022) is important for embodied agentTellex et al. (2011); Jansen (2020); Ahn et al. (2022) and conversational assistants (Ilievski et al., 2018; Yang et al., 2022). Previous work views procedural script learning as a structured form of commonsense knowledge Gupta et al. (2004); Regneri et al. (2010); Wanzare et al. (2016), while more recent work strengthens its association with the changing environments for executable action planning Puig et al. (2018); Shridhar et al. (2020). Some works (Sun et al., 2020; Zhao et al., 2021) explore to utilize human written programs to precisely specify tasks. Our method tackles the problem with aware of cause-effect by utilizing commonsense-infused prompts via a neuro-symbolic approach (Mao et al., 2019; Nye et al., 2021; Yi et al., 2018) for zero-shot procedural planning.

**Causality for Language Generation** The integration of causality and machine learning has been an intriguing topic for many problems Pearl (2009); Schölkopf (2022). Previous studies focusing on causal inference for natural language understanding Chen et al. (2020); Keith et al. (2020); Wood-Doughty et al. (2018) and generating counterfactual text representations Feder et al. (2021). Weber et al. (2020) proposes an intervention method for script learning. However, these methods cannot be directly applied to procedural planning which requires a formal structure. Our method is based on mediation analysis VanderWeele (2015) and causal intervention Pearl (2009); Peters et al. (2017).

**Prompt for Large Language Model** There is an emerging interest in using prompts to extract knowledge from large language models (Chen et al., 2022; Le Scao & Rush, 2021; Su et al., 2022; Ye et al., 2022; Zhou et al., 2022; Kojima et al., 2022). Cao et al. (2022) treats the prompt as a cause of the task-specific predictor and investigates biases in prompt-based probing evaluations. Chain of thought Wei et al. (2022) discovers that LLM can perform better on reasoning tasks when the prompt is designed as a series of short sentences that mimic the reasoning process of humans.

## 6 Conclusion and Future Work

Procedural planning is a newly emerged research area of great importance to various applications, such as household robots and virtual assistants. We propose a neuro-symbolic procedural **PLAN**ner (PLAN) with commonsense-infused prompts elicited from the external knowledge base to solve the procedural planning problem in a zero-shot manner without human annotated exemplars. Experiments show the effectiveness of our proposed PLAN under both origin and counterfactual settings, indicating the capability of mitigating spurious correlation by injecting external knowledge in LLMs. Though, procedural planning over long-horizon and composite tasks remains challenging. And exploring multimodal learning and developing human-aligned evaluation metrics are promising future directions in this area.

## 7 ETHICAL STATEMENT

Given the limited diversified cultural background of the dataset we are using from RobotHow and WikiHow, we assume our results may be biased toward a single cultural background. For instance, given the task "make breakfeast", it should take multi-culture into consideration to generate the procedural plans.

## 8 REPRODUCIBILITY STATEMENT

We provide more data samples and qualitative samples in supplemental materials. In addition, we provide our code implementation at https://anonymous.4open.science/r/PLANNER-7B24 to reproduce our experiments. The `Preprocess` folder provides the utils to construct the data. The `Evaluation` folder provides the code for automatic and human evaluation tools. The `Planning` folder contains the main code for our approach and reproduced planners for procedural planning. The `Visualization` folder provides the code we use to visualize in the environment.

## ACKNOWLEDGMENTS

The research was sponsored by the U.S. Army Research Office and was accomplished under Contract Number W911NF-19-D-0001 for the Institute for Collaborative Biotechnologies. This work was also supported by the National Science Foundation award #2048122. We thank the Robert N.Noyce Trust for their generous gift to the University of California via the Noyce initiative. The views and conclusions contained in this document are those of the authors and should not be interpreted as representing the official policies, either expressed or implied, of the U.S. Government. The U.S. Government is authorized to reproduce and distribute reprints for Government purposes notwithstanding any copyright notation herein.

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

# Appendix

## Table of Contents

# A   SCM THEORETICAL DETAILS

## A.1   CAUSAL PRELIMINARIES

The Structural Causal Model (SCM) is a directed acyclic graph (DAG) to describe the causal relationships within a system Pearl (2009). In this paper, we refer to the unrolled SCM along the time dimension as the full temporal causal graph, while the rolled-up version is also called the causal summary graph Peters et al. (2017). In an SCM, if the variable $D$ is a cause of both $T$ and $S_i$, then it is called a **confounder**. A confounder opens up a backdoor path and causes a spurious correlation between $T$ and $S_i$. The **backdoor path** is defined as the remaining path between $T$ and $S_i$ when all the arrows pointing out of $T$ are removed. Therefore, $T \leftarrow D \rightarrow S_i$ is a backdoor path. For our SCM with mediator $P_i$ shown in Figure 4c (same as Figure 2b) from the main paper, there is no backdoor path between $T$ and $\{P_i, S_{i-1}\}$ because only $D \rightarrow T$ is left after removing outgoing arrows of $T$. On the other hand, there is a backdoor path between $P_i$ and $S_i$, i.e. $P_i \leftarrow T \leftarrow D \rightarrow S_i$ so that $P_i$ indirectly affects the observation of $S_i$ through $\{T, S_{i-1}\}$ and $D$. The **mediator** is the variable added between treatment variable (the cause $T$ and $S_{i-1}$ in our case) and treatment variable (the effect $S_i$ in our case), and thus blocks all directed path from the cause to effect ( (Zhang et al., 2016)). The **spurious correlations** happens when two variables are statistically related but not causally related because of a third variable influences these two variables at the same time or the correlation is coincidental.

To identify the true causal effect between $X$ and $Y$, we aim to estimate the conditional $\pi(Y|do(X))$ after intervention with the *do*-operator. The *do*-operator is to break the backdoor path by setting $X$ to a fixed value independent of $Z$. Then the path $Z \rightarrow X$ can be removed to eliminate the backdoor paths. In practice, the backdoor adjustment and front-door adjustment are two fundamental methods to implement interventions and obtain the conditional $\pi(Y|do(X))$.

**Clarity of the Definition** As a language prompt, $P_i$ inherits the content from $P_{i-1}$ and thus can be detached from steps before $S_{i-1}$ for simplicity.

**Causal Intervention** There are two types of operation to control the confounding bias: the *backdoor adjustment* and the *front-door adjustment* (Pearl, 2009). The backdoor adjustment is intractable in our case because it requires the prior distribution of the confounding variables. On the other hand, we can construct an input prompt as a mediator $P_i$ for $T \rightarrow S_i$ and $S_{i-1} \rightarrow S_i$. Then the front-door adjustment applies a two-step *do*-operation to mitigate bias by investigating $P \rightarrow S_i$ (Pearl, 2009). Specifically, we construct the prompt mediator $P_i$ using techniques illustrated in Section 2.2.

The pre-trained knowledge (D) in LLMs confounds language models to make biased decisions toward an unreasonable action. Since the confounder is unobservable, intervention techniques such as back-door (definition in Appendix A.2) adjustment (Hu & Li, 2021; Weber et al., 2020; Yue et al., 2020) are not applicable in our SCM. Instead, we build a mediator and implement it as a commonsense-infused prompt. Through the mediator, we can identify causal effects among goals and steps by investigating the indirect effect from the goals, which is essentially the front-door adjustment (definition in Appendix A.3) in causality (Pearl, 2009).

## A.2   THE BACKDOOR ADJUSTMENT

The backdoor adjustment is one way to realize the intervention $do(T = t)$ by considering the conditional probability over the existing data distribution with observed confounder $D$. Let $\pi_i$ denote $\pi(\cdot|P_{i-1})$ that represent the probability density function conditioned on $P_{i-1}$. It calculates the average causal effects by considering all stratums of the dataset:

$$\pi_i(S_i|do(T)) = \sum_d \pi_i(S_i|T, D = d)\pi_i(D = d) \tag{5}$$

However, for LLMs, the pretraining data is usually unobservable and has been transformed as knowledge incorporated into the hidden space. Therefore, we are not able to directly apply the backdoor adjustment.

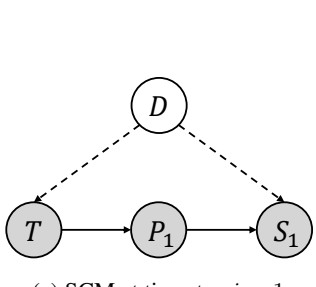
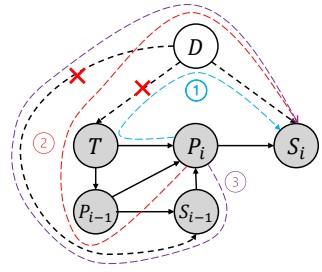
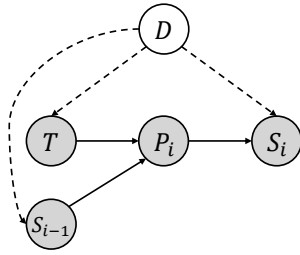

(a) SCM at timestep $i = 1$

(b) The SCM and three backdoor paths at timestep $i > 1$

(c) Equivalent SCM at timestep $i > 1$ after eliminating $P_{i-1}$

Figure 4: **The front-door Adjustment for Causal Procedural Planner.** (a) the structural causal model at timestamp $i = 1$. $T$ denotes the task name and $S_1$ denotes the step at timestep 1. $D$ is the unobservable confounding variable introduced by the pre-training data. $P_1$ denotes the mediating variables we construct to mitigate the spurious correlation at timestep 1. (b) $D$ opens up backdoor paths for $T \to S_i$, $P_{i-1} \to S_i$ and $S_{i-1} \to S_i$ which can be blocked by introducing $P_i$. path 1 and path 2 share the same path $D \to T$. Intervention on $T$ blocks $D \to T$ and the backdoor path 2. Intervention on $S_{i-1}$ blocks $D \to S_{i-1}$ and the backdoor path 3. (c) the structural causal model at timestamp $i > 1$ after simplification based on Equation 12-16.

### A.3 THE FRONT-DOOR ADJUSTMENT

The front-door adjustment is another technique to apply intervention by introducing a mediator $P_i$ when the confounder is unobservable. As is explained in Section 2.2 from the main paper, the front-door adjustment is equivalent to two consecutive *do*-operations on task $T$ and prompt $P_i$. We first investigate the generation of $S_1$ and then expand it to $S_t$.

**Timestep $i = 1$**   As is shown in Figure 4a, since there is no preceding steps, the first step generation involves $D$, $T$ and $P_1$ only. Similar to the proof in Section 2.2 from the main paper, we have:

$$
\begin{aligned}
\pi_i(S_1|do(T)) &= \sum_p \pi_i(S_1|do(P_1 = p))\pi_i(p|do(T)) \\
&= \sum_p \pi_i(p|T) \sum_t \pi_i(S_i|p, T = t)\pi_i(T = t)
\end{aligned}
\tag{6}
$$

By adding intervention to $T$, we make the value of $do(T = t)$ independent of the confounder $D$ at the beginning. The backdoor path through $D \to T$ is eliminated as a result.

**Timestep $i > 1$**   As is shown in Figure 2a from the main paper, we model the mediator $P_1$ as an effect of three variables, $T$, $P_{i-1}$ and $S_{i-1}$. The first step of our front-door adjustment is to apply the *do*-operator on the three variables and observe the change in $P_i$ as explained in Section 2.2 from the main paper. Since there are no backdoor paths between $P_i$ and these variables, we have the probability after intervention equal to the conditional probability without intervention:

$$
\pi_i(P_i = p|do(T)) = \pi_i(P_i = p|T)
\tag{7}
$$
$$
\pi_i(P_i = p|do(P_{i-1})) = \pi_i(P_i = p|P_{i-1})
\tag{8}
$$
$$
\pi_i(P_i = p|do(S_{i-1})) = \pi_i(P_i = p|S_{i-1})
\tag{9}
$$

The second step is to apply *do*-operator on $P_i$ and then identify the causal effect as:

$$
\begin{aligned}
\pi_i(S_i|do(P_i)) = \sum_{t,p',s} \Big( &\pi_i(S_i|P_i, T = t, P_{i-1} = p', S_{i-1} = s) \\
&\pi_i(T = t, P_{i-1} = p', S_{i-1} = s)\Big)
\end{aligned}
\tag{10}
$$

Combining Equation 7-9 and Equation 10, we have the front-door adjustment. Note that there are three backdoor paths from each of the variables $T$, $P_{i-1}$, and $S_{i-1}$, as is shown in Figure 4b (drawn

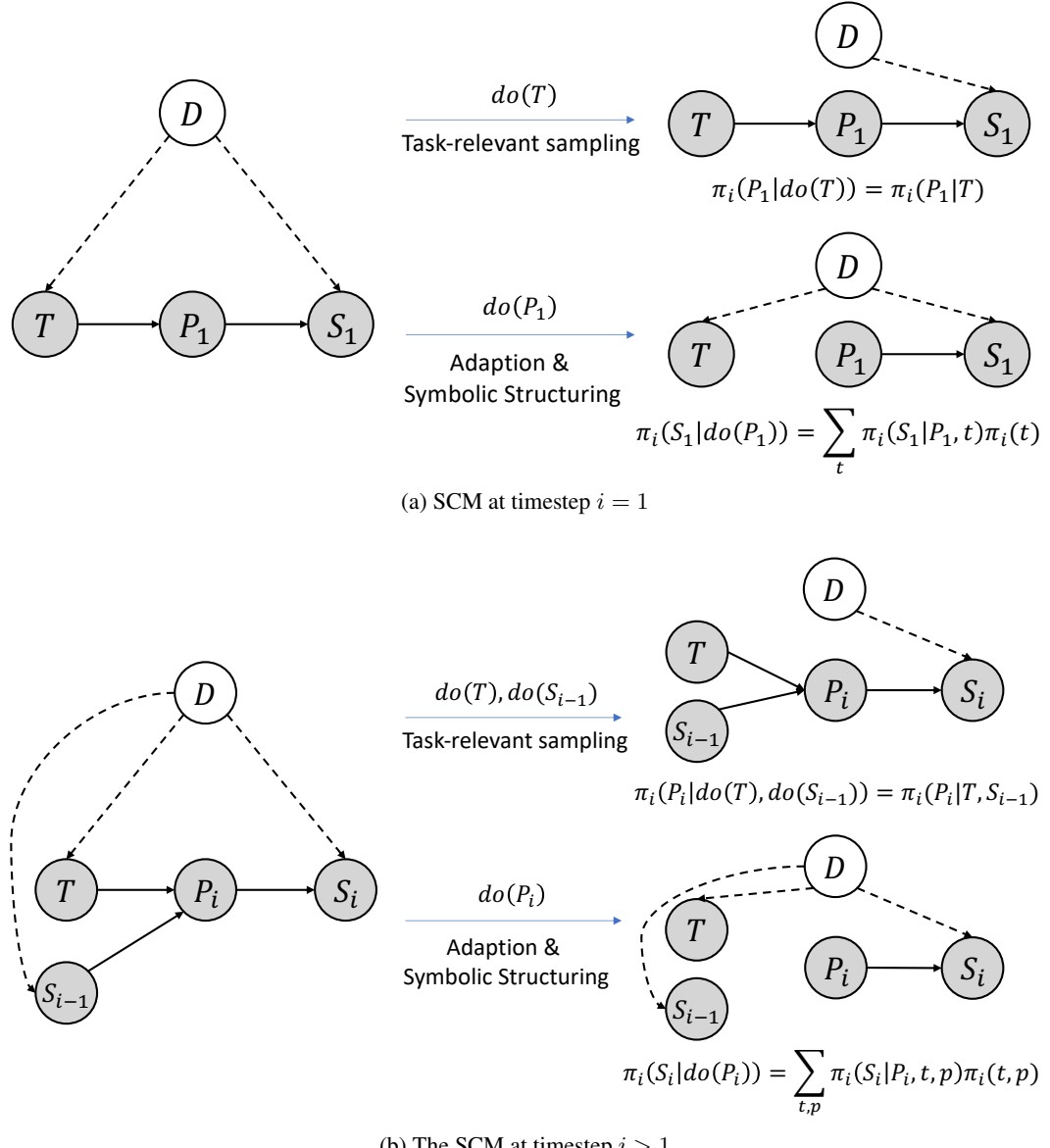

(a) SCM at timestep $i = 1$

(b) The SCM at timestep $i > 1$

Figure 5: **The Causal Graph after $do$-operation.** (a) the causal graph transition of Structural Causal Model at timestamp $i = 1$. (b) the causal graph transition of Structural Causal Model at timestamp $i > 1$.

in blue, red and purple). More importantly, the one through $T$, i.e. $P_i \leftarrow T \leftarrow D \rightarrow S_i$ (the blue path in Figure 4b) and the one through $P_{i-1}$, i.e. $P_i \leftarrow P_{i-1} \leftarrow T \leftarrow D \rightarrow S_i$ (the red path in Figure 4b) shares the same subpath. The intervention on the task $T$ breaks the backdoor paths for both $T$ and $P_{i-1}$. Therefore, we have our front-door adjustment as

$$\pi_i(S_i|do(S_{i-1}),do(P_{i-1}),do(T)) \tag{11}$$

$$= \sum_p \pi_i(S_i|do(P_i = p))\pi_i(p|do(S_{i-1}),do(P_{i-1}),do(T)) \tag{12}$$

$$= \sum_p \pi_i(S_i|do(P_i = p))\pi_i(p|do(S_{i-1}),P_{i-1},do(T)) \tag{13}$$

$$= \sum_p \pi_i(S_i|do(P_i = p))\pi_i(p|do(S_{i-1}),do(T)) \tag{14}$$

$$= \sum_p \pi_i(p|S_{i-1},T) \sum_{s,t} \pi_i(S_i|p,S_{i-1} = s,T = t)\pi_i(S_{i-1} = s,T = t) \tag{15}$$

$$= \pi_i(S_i|do(S_{i-1}),do(T)) \tag{16}$$

We have Equation 13 because of the intervention on $T$ and **Rule 2** (Pearl, 1995), Equation 14 because of **Rule 1** (Pearl, 1995). After simplification based on Equation 12-16, we get the SCM at timestep $i > 1$ in Figure 4c. This is an equivalent SCM after eliminating $P_{i-1}$ in Figure 4b. The reason we could eliminate $P_{i-1}$ is as follows. We follow a common method of constructing temporally-extended prompt, which is to append the prediction at previous timesteps to the prompt at current timestep. In our case, the $P_{G,i}$ is the same as $P_{G,i-1}$, thus $P_i$ inherit part of the content from $P_{i-1}$, the change only depend on the $S_{i-1}$. Thus $P_{i-1}$ and $S_{i-2}$ are fixed, and there is no need to predict $P_{i-1}$ at timestep $i$ again. In this way, we simplify the causal graph in Figure 4b to the one in Figure 4c. In summary, we define and simplify the causal graph based on the temporal-extended property of our prompt construction ($P_i$ inherit the content from $P_{i-1}$). We end up with Equation 14-16 which is shown as Equation 3 in Section 2.2 from the main paper.

## B   IMPLEMENTATION DETAILS

### B.1   ORIGINAL DATASET DETAILS

**RobotHow** This dataset is Attribution-NonCommercial-ShareAlike 4.0 International Creative Commons License. We evaluate the inference of 150 tasks by random selection from the dataset. Each program contains the task name, task description and steps. We use the task name and sequence of steps as our input and output references.Each step is a composition of `[Action]`, `[Object]` and `[Number]`. For example, the sequence of steps of the task "Watch TV" are: 1. [Walk] <TELEVISION> (1) 2. [SwitchOn] <TELEVISION> (1) 3. [Walk] <SOFA> (1) 4. [Sit] <SOFA> (1) 5. [Watch] <TELEVISION> (1).

**WikiHow** This dataset[2] is under an Attribution-Noncommercial-Share Alike 3.0 Creative Commons License. And the text content is free to modify, republish and share. We evaluate the inference of 1000 tasks by random selection from the dataset. The admissible action space and interaction object space are more complex than the programs in RobotHow. And there is no fixed "[Action] ¡Object¿ (Number)" form of each step. For each article, it contains the title, the bold headlines and text. We utilize the title and headlines as our task name and steps respectively.

**External Knowledge Base** For the external knowledge base, we utilize ConceptNet to leverage commonsense reasoning ability to help ground language generation in goal-guided procedural text generation. ConceptNet (Speer et al., 2017) captures commonsense knowledge explicitly with triplets of (*head node*, *relation*, *end node*). It contains $799,273$ nodes and $2,487,810$ edges that represent both symmetric and asymmetric relations. Specifically, the core relations we utilized are *Synonym*, *AtLocation*, *CapableOf*, *Causes*, *CausesDesire*, *HasPrerequisite*, *HasSubevent*, and *UsedFor*. Since we are looking at the commonsense knowledge in house-holding tasks, so we filter out the relations (/r/DistinctFrom, /r/DerivedFrom, /r/SymbolOf, /r/EtymologicallyRelatedTo, /r/EtymologicallyDerivedFrom) that are related to the linguistic.

---

[2]https://www.wikihow.com

## B.2 COUNTERFACTUAL DATASET AND EXPERIMENT DETAILS

| Category | Description | Example |
|---|---|---|
| Initial Configuration | Constrain the environment configuration, e.g. location | Watch TV in bedroom |
| Intermediate Step | Constrain the way to finish the task | Work (Find Computer) |
| Final Goal | Change the final effect of the task by composition | Watch youtube and Put away jackets |

Table 5: **Three Types of Counterfactual Procedural Planning.** Three types of methods, including initial configuration, intermediate step, and final goal are applied to intervene the original procedural data.

| Category | Original Program | Counterfactual Program |
|---|---|---|
| Initial Configuration | Task: Watch TV
Step 1: Find remote control.Step 2: Grab remote control.
Step 3: Find television.Step 4: Switch on television.
Step 5: Turn to television.Step 6: Watch television.
Step 7: Switch off television.
Step 8: Put back remote control | Task: Watch TV in bedroom
Step 1: Walk to bedroom Step 2: Find remote control.
Step 3: Grab remote control.Step 4: Find television.
Step 5: Switch on television.Step 6: Turn to television.
Step 7: Watch television.Step 8: Switch off television.
Step 9: Put back remote control |
| Intermediate Step | Task: Work
Step 1: Walk to home office.Step 2: Walk to chair.
Step 3: Find chair.Step 4: Sit on chair.
Step 5: Find computer.Step 6: Switch on computer.
Step 7: Turn to computer.Step 8: Look at computer | Task: Work (Find Computer)
Step 1: Walk to home office.Step 2: Walk to chair.
Step 3: Find chair.Step 4: Sit on chair.
Step 5: Find computer.Step 6: Switch on computer.
Step 7: Turn to computer.Step 8: Look at computer |
| Final Goal | Task1: Turn light off
Step 1: Walk to bedroom Step 2: Walk to light
Step 3: Switch off light
Task2: Clean
Step 1: Walk to home office Step 2: Walk to rag
Step 3: Find rag Step 4: Grab rag
Step 5: Walk to desk Step 6: Find computer
Step 7: Wipe computer Step 8: Wipe desk
Step 9: Put back rag | Task: Turn light off and Clean
Step 1: Walk to bedroom Step 2: Walk to light
Step 3: Switch off light
Step 4: Walk to home office
Step 5: Walk to rag Step 6: Find rag
Step 7: Grab rag Step 8: Walk to desk
Step 9: Find computer Step 10: Wipe computer
Step 11: Wipe desk Step 12: Put back rag |

Table 6: **Comparison between Standard and Counterfactual Procedural Planning.** Three types of methods, including initial configuration, intermediate step, and final goal are applied to intervene the original procedural data.

Table 6 show the examples that compare the original program and the counterfactual program of each intervention method are also provided. Specifically, for **Initial Configuration**, we randomly append the location to a given task name to constrain the location of completing the task. The steps are prepended with the initial step "walk to ¡Location¿". For **Intermediate Step**, we randomly sampled a step from the task-specific program and append it to the task name to constrain the way to implement a given task. For **Final Goal**, we randomly combine two tasks by combining both the task names and the programs to construct a set of long-horizon composite tasks.

We conduct counterfactual experiments by applying randomly selected intervention methods over RobotHow. And we only apply the Intermediate Step intervention method over WikiHow due to the loose configuration requirement and the long text of the WikiHow contents. Note that the performance gain of PLAN under the counterfactual setting mainly comes from the additional guidance of the task introduced from the Intermediate Step intervention method. However, the baselines mostly experience performance drops due to the limited annotated exemplars. PLAN consistently outperforms baselines by a large margin, indicating its superiority under the counterfactual setting.

## B.3 METHOD DETAILS

The existing formalization of the procedural planning task can be mainly categorized as 1) sequential choice making (Lyu et al., 2021; Wu et al., 2022; Zhang et al., 2020a;b), which reasons about the next step from the options given, the task, and previous steps; 2) conditioned generation (Huang et al., 2022; Ahn et al., 2022), which generates the temporally extended plans to implement the task. We study the procedural planning task as the conditioned generation problem (Huang et al., 2022; Ahn et al., 2022) since it resembles real-world scenarios.

**Baselines** LLMaP propose a procedure to extract temporally extended plans from large pre-trained language models. Chain explores manually creating exemplars that mimic the reasoning process

and uses them to prompt large language models for reasoning tasks. To compare with Chain on the procedural planning task, we manually generate exemplars that contain the chain of thought for $1\%$ of the inference task programs. Note that for the BART language model, we use BART-large version. And we use the $1.5$ billion parameter GPT-2 (aka gpt2-xl). For the translation model $LM_T$, we use sentence-transformers (RoBERTa-large). All these models are released by HuggingFace. In addition, our experiments with GPT3 (davinci) use OpenAI API (May, 2022).

**External Knowledge Graph** Conceptnet5 define a set of $34$ relations ([3]). Within the relations we consider in the procedural planning task, the averaged sampling time of subgraph sampling is 0.03576 milliseconds per task program.

### B.4   HYPERPARAMETER SEARCH AND CONFIGURATION DEICISION

We perform a hyperparameter search for all evaluated methods for the following hyperparameters.

- The confidence threshold $\theta$, which terminate the generation when below it, is searched in $\{0, 0.5, 0.55, 0.6, 0.65, 0.7, 0.75, 0.8\}$.
- The steps horizon, which constrains the maximal number of procedural planning steps, is searched in $\{10, 20, 40\}$.
- The number of hops for retrieving the subgraph from the external knowledge base is searched in $\{1, 2, 3\}$.
- The ratio of maximal concepts to the length of the task name is searched in $\{1, 2, 3\}$.
- The cosine similarity threshold for keeping the task-specific concept is searched in $\{0.4, 0.6, 0.8\}$.
- The edge weight threshold $\theta_e$ is searched in $\{0.1, 0.2, 0.3, 0.4, 0.5, 0.6, 0.7, 0.8\}$.
- The top-$k$ task-specific nodes value is searched in $\{1, 5, 10, 15, 20, 25, 50, 100\}$.

The configurations used in the experiments are: $\theta$=0.7, 20 step horizon, 3 hops, 3 ratio of concepts to task length, cosine similarity threshold $0.4$, $\theta_e$=0.6 and $k$=10.

We empirically choose the hop number $H$ as 3 considering both the input length limit of the LLMs and the fact that 3-hop contains reasonable relevant information in practice (Zhang et al., 2022).

### B.5   COMPUTATION AND RESOURCES

We use one single NVIDIA A100 GPU Server for all the experiments. Since there is no training in our zero-shot settings, the computation is only used for the inference stage of the experiments.

## C   EVALUATION DETAILS

### C.1   CROWDSOURCING HUMAN EVALUATION

We conduct all the human evaluations (rating and win-lose comparison) on Amazon Mechanical Turk platform. Each example is rated by 3 annotators. We ask Amazon Mechanical Turk workers, for every assignment, to evaluate the quality of the provided low-level steps given the high-level task description. For the **Win-Lose Comparison**, they were asked to choose one from the two provided model generated results by *1:the first one is better*, *2:equal* and *3:the second one is better*. For the **Human Ratings**, they were asked to score each sample with 5-point Likert scale. This process does not involve collecting any personal information. And we manually check no offensive content is produced by the models.

The assignment layout templates for workers are shown in Figure 7 and Figure 6. Specifically, we evaluate randomly selected 50 task examples from each dataset (RobotHow and WikiHow) under all the settings (standard and counterfactual). We only collect the examples that the workers read the instructions carefully by checking whether they give 1 score for the empty program as a sanity check. The hourly wage paid to participants is estimated $9. And the total amount spent on participant

---

[3]https://github.com/commonsense/conceptnet5/wiki/Relations

compensation is $1296. The details of the Human Intelligence Tasks process are described in the following sections.

Figure 6: Amazon Mechanical Turk Platform. Questions Layout for Human Raters for Win-Tie-Lose Comparison.

### C.1.1 WIN-LOSE COMPARISON

During the process of Human Intelligence Tasks, the workers are shown the following instructions: *Read the given task and the sequence of steps, determine which set of steps can better complete the target task. In other words, can the task be decomposed into these steps? Please consider the sequential order of the steps.*

Then the program to be evaluated is provided as:

**Question** Task: Study

**Sequence 1:**: Step 1: Walk to textbook Step 2: Read book Step 3: Walk to book

**Sequence 2:**: Step 1: Walk to home office Step 2: Find desk

Finally, the workers are asked to score the program by following the instructions below: *Select an option: 1 - Sequence 1 is better; 2 - Tie; 3 - Sequence 2 is better*

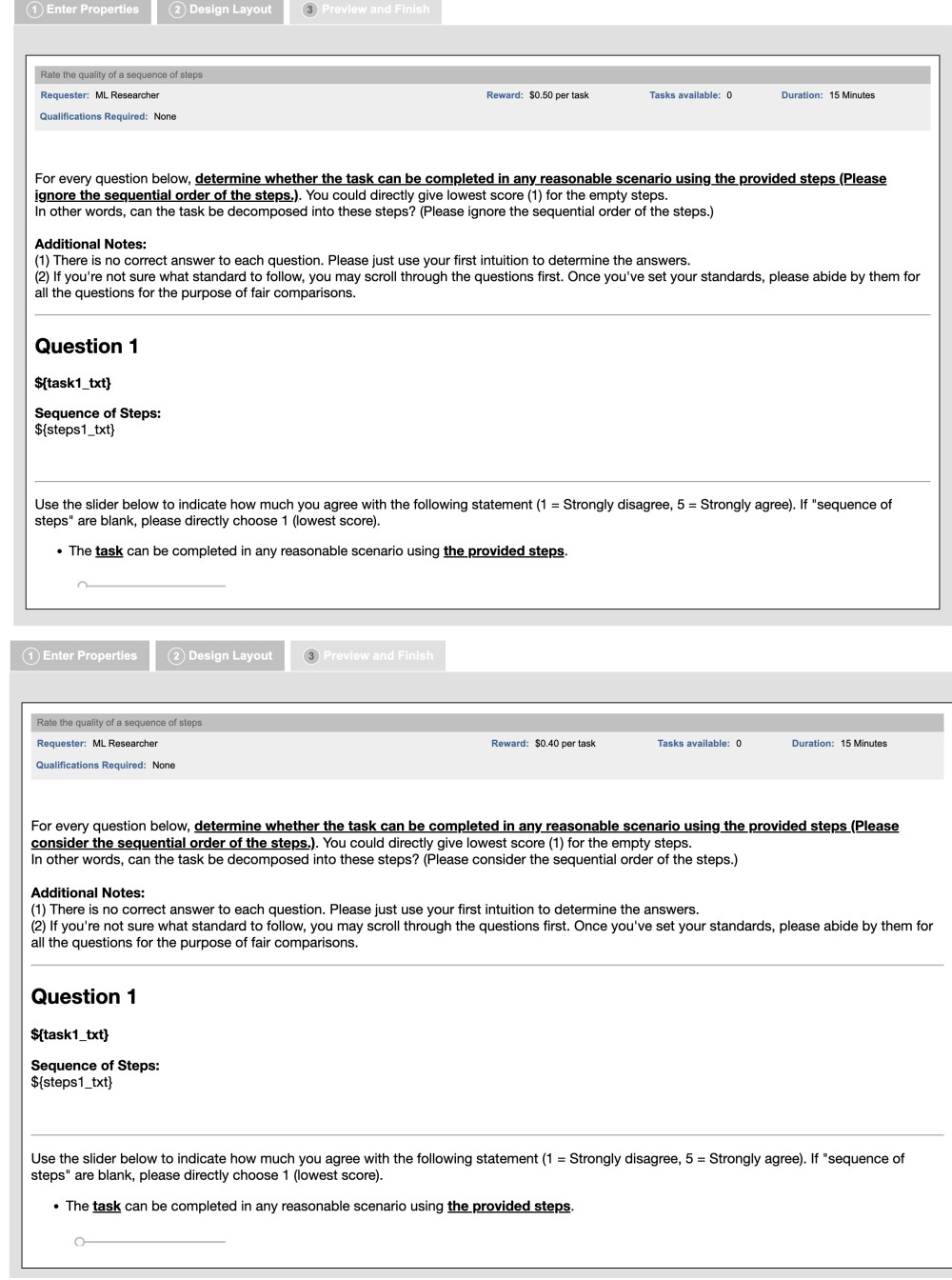

Figure 7: Amazon Mechanical Turk Platform. Questions Layout for Human Raters for 5 Point Likert Scale.

The above example is to evaluate the order metric, for the coverage metric, the same process are conducted, except for the instructions are: *Read the given task and the sequence of steps, and determine which sequence covers more steps that are necessary to complete the target task. Please ignore the sequential order of the steps.*

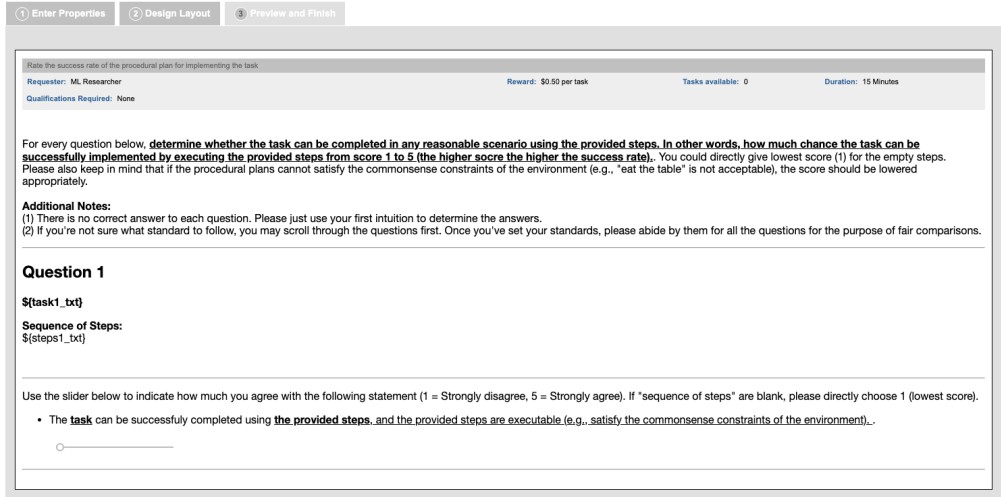

Figure 8: Amazon Mechanical Turk Platform. Questions Layout for Human Raters for 5 Point Likert Scale on Success Rate.

### C.1.2 HUMAN RATINGS

Similar as the Win-Lose Comparison Human Intelligence Tasks, the workers are shown the following instructions: *For every question below, determine whether the task can be completed in any reasonable scenario using the provided steps (Please consider the sequential order of the steps.). You could directly give the lowest score (1) for the empty steps. In other words, can the task be decomposed into these steps? (Please consider the sequential order of the steps.)*

Then the program to be evaluated is provided as:

**Question** Task: Write an email

**Sequence of Steps**: Step 1: Walk to home office Step 2: Walk to computer Step 3: Find computer Step 4: Turn to computer Step 5: Look at computer Step 6: Walk to computer Step 7: Find chair Step 8: Sit on chair Step 9: Find keyboard Step 10: Grab keyboard Step 11: Find mouse Step 12: Grab mouse Step 13: Type on keyboard

Finally, the workers are asked to score the program by following the instructions below: *Use the slider below to indicate how much you agree with the following statement (1 = Strongly disagree, 5 = Strongly agree). If "sequence of steps" are blank, please directly choose 1 (lowest score). The task can be completed in any reasonable scenario using the provided steps.* [SLIDER PROVIDED HERE]

The above example is to evaluate the order metric, for the coverage metric, the same process is conducted, except for the instructions are: *For every question below, determine whether the task can be completed in any reasonable scenario using the provided steps (Please ignore the sequential order of the steps.). You could directly give the lowest score (1) for the empty steps. In other words, can the task be decomposed into these steps? (Please ignore the sequential order of the steps.)*

### C.2 MORE RESULTS

**Significance Test** We provide paired-t test ($p<0.05$) statistics results for Table 2. On RobotHow, our PLAN significantly outperforms all baselines on Original-Order(BART) and Counterfactual-Coverage(GPT2). On WikiHow, our PLAN significantly outperforms all baselines on Original-Coverage(BART, GPT2), Counterfactual-Coverage(BART, GPT2), and Counterfactual-Order(BART). For the coverage metric under the counterfactual setting, the human-provided program is not significantly better than our PLAN.

We also conduct the paired-t test ($p<0.05$) statistics results over the variant "$w/o$ Adaption" and "$w/o$ Symbolic". Compared with the full model PLAN, the variants experienced a statistically significant

| Model | RobotHow | | | | | | | | |
|---|---|---|---|---|---|---|---|---|---|
| | Step Bucket | S-BLEU | WMD | BERT-f1 | ROUGE-f1 | Coverage | Order | Step Avg. | Time Cost (ms) |
| BART + Chain (Wei et al., 2022) | (0,10] | 0.073 | 0.915 | 0.863 | 0.432 | 2.947 | 2.760 | 6.600 | 3.330 |
| | (10,20] | 0.049 | 0.909 | 0.857 | 0.442 | 2.921 | 2.825 | 13.714 | 3.820 |
| BART + LLMaP (Huang et al., 2022) | (0,10] | 0.076 | 0.941 | 0.867 | 0.450 | 2.973 | 2.760 | 6.600 | 3.298 |
| | (10,20] | 0.028 | 0.931 | 0.853 | 0.393 | 3.095 | 2.889 | 13.714 | 3.866 |
| BART + PLAN | (0,10] | 0.099 | 0.955 | 0.894 | 0.532 | 3.187 | 3.013 | 6.600 | 3.272 |
| | (10,20] | 0.041 | 0.935 | 0.869 | 0.437 | 3.079 | 3.206 | 13.714 | 4.022 |
| GPT2 + Chain (Wei et al., 2022) | (0,10] | 0.076 | 0.891 | 0.856 | 0.395 | 2.453 | 2.147 | 6.600 | 3.370 |
| | (10,20] | 0.057 | 0.877 | 0.845 | 0.359 | 2.365 | 2.476 | 13.714 | 3.804 |
| GPT2 + LLMaP (Huang et al., 2022) | (0,10] | 0.112 | 0.942 | 0.894 | 0.486 | 3.147 | 2.987 | 6.600 | 3.212 |
| | (10,20] | 0.064 | 0.906 | 0.859 | 0.394 | 2.921 | 2.73 | 13.714 | 3.875 |
| GPT2 + PLAN | (0,10] | 0.167 | 0.940 | 0.901 | 0.554 | 3.173 | 2.813 | 6.600 | 3.344 |
| | (10,20] | 0.101 | 0.924 | 0.882 | 0.480 | 2.984 | 3.063 | 13.714 | 3.954 |
| GPT3 + Chain (Wei et al., 2022) | (0,10] | 0.086 | 0.920 | 0.878 | 0.445 | 3.568 | 3.459 | 6.838 | 3.215 |
| | (10,20] | 0.112 | 0.931 | 0.884 | 0.499 | 3.562 | 3.469 | 13.688 | 3.988 |
| GPT3 + LLMaP (Huang et al., 2022) | (0,10] | 0.132 | 0.951 | 0.911 | 0.544 | 3.811 | 3.486 | 6.838 | 3.144 |
| | (10,20] | 0.139 | 0.939 | 0.894 | 0.502 | 3.531 | 3.625 | 13.688 | 3.964 |
| GPT3 + PLAN | (0,10] | 0.171 | 0.961 | 0.918 | 0.574 | 3.459 | 3.568 | 6.838 | 3.379 |
| | (10,20] | 0.167 | 0.953 | 0.916 | 0.578 | 3.750 | 3.688 | 13.688 | 4.134 |

Table 7: Evaluation results on the Original RobotHow by separating test set into several Step Bucket.

| Model | WikiHow | | | | | | | | |
|---|---|---|---|---|---|---|---|---|---|
| | Step Bucket | S-BLEU | WMD | BERT-f1 | ROUGE-f1 | Coverage | Order | Step Avg. | Time Cost (ms) |
| BART + Chain (Wei et al., 2022) | (0,10] | 0.053 | 0.919 | 0.789 | 0.356 | 2.969 | 3.521 | 6.156 | 8.233 |
| | (10,20] | 0.032 | 0.921 | 0.784 | 0.294 | 2.644 | 3.311 | 14.467 | 7.349 |
| BART + LLMaP (Huang et al., 2022) | (0,10] | 0.068 | 0.934 | 0.814 | 0.353 | 2.802 | 3.438 | 6.156 | 8.289 |
| | (10,20] | 0.032 | 0.924 | 0.794 | 0.293 | 2.600 | 3.178 | 14.467 | 7.487 |
| BART + PLAN | (0,10] | 0.108 | 0.939 | 0.834 | 0.431 | 3.083 | 3.594 | 6.156 | 8.341 |
| | (10,20] | 0.059 | 0.927 | 0.812 | 0.372 | 2.978 | 3.244 | 14.467 | 7.829 |
| GPT3 + Chain (Wei et al., 2022) | (0,10] | 0.107 | 0.928 | 0.817 | 0.353 | 3.031 | 3.438 | 6.156 | 8.367 |
| | (10,20] | 0.077 | 0.933 | 0.812 | 0.328 | 2.733 | 3.422 | 14.467 | 7.585 |
| GPT3 + LLMaP (Huang et al., 2022) | (0,10] | 0.111 | 0.946 | 0.831 | 0.36 | 3.292 | 3.625 | 6.156 | 8.218 |
| | (10,20] | 0.066 | 0.955 | 0.829 | 0.342 | 2.978 | 3.378 | 14.467 | 7.583 |
| GPT3 + PLAN | (0,10] | 0.136 | 0.961 | 0.856 | 0.416 | 3.645 | 4.0 | 6.677 | 8.213 |
| | (10,20] | 0.127 | 0.961 | 0.868 | 0.458 | 3.68 | 3.2 | 13.6 | 7.632 |
| GPT3 + Chain (Wei et al., 2022) | (0,10] | 0.123 | 0.954 | 0.837 | 0.432 | 3.655 | 3.517 | 6.0 | 8.424 |
| | (10,20] | 0.121 | 0.949 | 0.856 | 0.465 | 3.421 | 3.684 | 15.526 | 7.775 |
| GPT3 + LLMaP (Huang et al., 2022) | (0,10] | 0.146 | 0.956 | 0.865 | 0.514 | 3.652 | 3.739 | 6.565 | 7.953 |
| | (10,20] | 0.099 | 0.951 | 0.849 | 0.452 | 3.375 | 3.312 | 15.0 | 7.247 |
| GPT3 + PLAN | (0,10] | 0.203 | 0.969 | 0.861 | 0.506 | 3.31 | 3.643 | 5.81 | 8.101 |
| | (10,20] | 0.185 | 0.967 | 0.855 | 0.466 | 3.714 | 3.333 | 15.095 | 7.506 |

Table 8: Evaluation results on the Original WikiHow by separating test set into several Step Bucket.

performance drop. Especially on BERTScore-f1, the p-value is $8.884e-13$ and $1.4e-8$ respectively. This further confirms the importance of the modules.

**Results on GPT-3**   In addition, we conduct experiments with GPT-3 (davinci version) using OpenAI API. We showcase the comparison in Table 9 and Table 10.

| Model | Program |
|---|---|
| | **RobotHow** Task: Write an Email |
| Human | Step 1: Walk to home office. Step 2: Walk to computer. Step 3: Find computer. Step 4: Turn to computer. Step 5: Look at computer. Step 6: Walk to computer. Step 7: Find chair. Step 8: Sit on chair. Step 9: Find keyboard. Step 10: Grab keyboard. Step 11: Find mouse. Step 12: Grab mouse. Step 13: Type on keyboard |
| Chain | Empty plan prediction due to low confidence for the first step. |
| LLMaP | Step 1: Switch on computer. |
| PLAN | Step 1: point at mail. Step 2: put envelope on printer.Step 3: put notes on printer. |
| | **RobotHow** Task: Turn on light |
| Human | Step 1: Walk to home office. Step 2: Walk to floor lamp. Step 3: Switch on floor lamp |
| Chain | Step 1: Switch on light switch. |
| LLMaP | Step 1: Switch on light switch. |
| PLAN | Step 1: Switch on light. Step 2: Switch on lamp. Step 3: Turn to light switch. Step 4: Switch on light switch. |
| | **RobotHow** Task: Drink |
| Human | Step 1: Find water glass. Step 2: Grab water glass. Step 3: Drink water glass. |
| Chain | Step 1: Find coffee cup. |
| LLMaP | Step 1: Find cup. Step 2: Drink cup. Step 3: Pour cup into glass. |
| PLAN | Step 1: Grab glass. Step 2: Pour water into drinking glass. Step 3: Drink water. |
| | **RobotHow** Task: Play games |
| Human | Step 1: Walk to bedroom. Step 2: Walk to video game console. Step 3: Find video game console. Step 4: Walk to video game console. Step 5: Switch on video game console. Step 6: Find television. Step 7: Switch on television. Step 8: Find video game controller. Step 9: Grab video game controller. Step 10: Find chair. Step 11: Walk to chair. Step 12: Sit on chair |
| Chain | Empty plan prediction due to low confidence for the first step. |
| LLMaP | Step 1: Switch on console. Step 2: Put console on dvd player. |
| PLAN | Step 1: Find video game console. Step 2: Grab video game controller. Step 3: Put console on boardgame. Step 4: Look at video game console. |
| | **RobotHow** Task: Put groceries in Fridge |
| Human | Step 1: Walk to dining room. Step 2: Walk to food food. Step 3: Find food food. Step 4: Grab food food. Step 5: Find freezer. Step 6: Open freezer. Step 7: Drop food food |
| Chain | Step 1: Walk to fridge. Step 2: Find groceries. Step 3: Find fridge. Step 4: Open fridge. Step 5: Put groceries in fridge. Step 6: Close fridge. |
| LLMaP | Step 1: Open fridge. Step 2: Pull groceries. Step 3: Put groceries on blender. Step 4: Close fridge. |
| PLAN | Step 1: Walk to fridge. Step 2: Open fridge. Step 3: Put groceries in fridge. Step 4: Close fridge. |
| | **RobotHow** Task: Admire art |
| Human | Step 1: Walk to home office. Step 2: Walk to drawing. Step 3: Turn to drawing. Step 4: Look at drawing. Step 5: Find drawing. Step 6: Turn to drawing. Step 7: Look at drawing. |
| Chain | Step 1: Walk to drawing. |
| LLMaP | Step 1: Look at painting. |
| PLAN | Step 1: Look at centerpiece. Step 2: Put centerpiece on music stand. Step 3: Point at painting. Step 4: Look at drawing. Step 5: Touch centerpiece. Step 6: Look at painting. Step 7: Put centerpiece on love seat. |
| | **RobotHow** Task: Greet guests |
| Human | Step 1: Walk to dining room.Step 2: Walk to child.Step 3: Find child. Step 4: Greet child.Step 5: Find woman.Step 6: Greet woman. Step 7: Find chair.Step 8: Sit on chair |
| Chain | Empty plan prediction due to low confidence for the first step. |
| LLMaP | Step 1: Open door. Step 2: Close door. |
| PLAN | Step 1: Walk to dining room. Step 2: Walk to entrance hall. Step 3: Greet woman. |

Table 9: Showcases of procedural steps predicted by different models with GPT3 as the base LLM on RobotHow.

| Model | Program |
|---|---|
| | **WikiHow** Task: How to Become an Art Investor |
| Human | Step 1: Start with some experience or interest in art. 
 Step 2: Understand the difference between art collectors, art investors and art speculators. 
 Step 3: Figure out what you are willing to pay for art, before going to an auction house. 
 Step 4: Pay attention to what schools of art are selling well, and which are down. 
 Step 5: Focus art investments on fine art paintings, rather than decorative art. 
 Step 6: Reach out to trusted auction houses and dealers when you are looking to buy art. 
 Step 7: Buy your investment art when you feel confident of its worth, its price and its ability to grow in value. 
 Step 8: Study how art is properly stored. 
 Step 9: Have your art investments appraised occasionally.Step 10: Consider renting out your art investments. 
 Step 11: Understand that selling an art investment can take time. |
| Chain | Step 1: Buy your investment art when you feel confident of its worth, its price and its ability to grow in value. |
| LLMaP | Step 1: Reach out to trusted auction houses and dealers when you are looking to buy art. |
| PLAN | Step 1: Figure out what you are willing to pay for art, before going to an auction house. 
 Step 2: Consider renting out your art investments. 
 Step 3: Buy your investment art when you feel confident of its worth, its price and its ability to grow in value. |
| | **WikiHow** Task: How to Be an Organized Artist |
| Human | Step 1: Make sure you know what is expected of you. Step 2: Stick to your topic. 
 Step 3: Don't try to be to be funny unless the scenario calls for it. 
 Step 4: Act naturally for the situation; talk, act and sit as your character would usually do in the circumstances. 
 Step 5: Participate. Step 6: Don't react to what others say as yourself, stay in character. 
 Step 7: Don't make anything violent or too crazy. Step 8: Relax and enjoy yourself. 
 Step 9: Be your character. Step 10: Play games that allow you to practice improvisation. |
| Chain | Step 1: First, you will need to make sure you have all of the materials listed below. 
 Step 2: Set a schedule. Step 3: Create a comfortable space. 
 Step 4: Take notes in journal or sketchbo. Step 5: Keep neat and tidy. Step 6: Take a break. |
| LLMaP | Step 1: Make plans. |
| PLAN | Step 1: Start with some experience or interest in art. Step 2: Put together a schedule and chart. 
 Step 3: Prepare to create your neopoprealist mural. Step 4: Organize your computer-based materials. 
 Step 5: Have a clear plan. 
 Step 6: Buy your investment art when you feel confident of its worth, its price and its ability to grow in value. 
 Step 7: Work on being the best you. |
| | **WikiHow** Task: How to Be Good at Improvisation |
| Human | Step 1: Keep related supplies in the same area. 
 Step 2: Make an effort to clean a dedicated workspace after every session. 
 Step 3: Place loose supplies in large, clearly visible containers. 
 Step 4: Use clotheslines and clips to hang sketches, photos, and reference material. 
 Step 5: Use every inch of the room for storage, especially vertical space. 
 Step 6: Use chalkboard paint to make space for drafting ideas right on the walls. 
 Step 7: Purchase a label maker to make your organization strategy semi-permanent. 
 Step 8: Make a habit of throwing out old, excess, or useless stuff each month. |
| Chain | Step 1: Play games that allow you to practice improvisatio. |
| LLMaP | Step 1: Don't overdo it. |
| PLAN | Step 1: Try the spontaneous approach. 
 Step 2: Express yourself creatively. Step 3: Play games that allow you to practice improvisatio. 
 Step 4: Do extracurricular activitie. |
| | **WikiHow** Task: How to Train a Parrot to Say Something |
| Human | Step 1: Decide what you want your parrot to say, but make it basic. 
 Step 2: If you want, you can make it say simple but funny things. 
 Step 3: You should go to a nice and quiet room. 
 Step 4: To start teaching it, repeat what you want it to say many times. 
 Step 5: If you DO get your parrot to say it correctly, then you've succeeded! |
| Chain | Step 1: Decide what you want your parrot to say, but make it basic. |
| LLMaP | Step 1: If you do get your parrot to say it correctly, then you've succeeded. |
| PLAN | Step 1: Decide what you want your parrot to say, but make it basic. 
 Step 2: If you do get your parrot to say it correctly, then you've succeeded. |

Table 10: Showcases of procedural steps predicted by different models with GPT3 as the base LLM on WikiHow.

| Model | RobotHow | | WikiHow | |
| --- | --- | --- | --- | --- |
| | Original-Executability | Counterfactual-Executability | Original-Executability | Counterfactual-Executability |
| Chain (Wei et al., 2022) | 3.16 | 3.60 | 3.32 | 3.58 |
| LLMaP (Huang et al., 2022) | 3.60 | 3.88 | 3.42 | 3.74 |
| PLAN (Ours) | **3.84** | **3.90** | **4.02** | **3.84** |

Table 11: Averaged 5-point Likert scale human evaluations on `Success Rate` aspect with GPT3 language model architecture.

| Model | RobotHow | | | | WikiHow | | | |
| --- | --- | --- | --- | --- | --- | --- | --- | --- |
| | S-BLEU | WMD | BERT-f1 | ROUGE-f1 | S-BLEU | WMD | BERT-f1 | ROUGE-f1 |
| GPT3 + PLAN (Ours) | **0.155** | **0.939** | **0.902** | **0.561** | **0.155** | **0.961** | **0.849** | **0.433** |
| $w/o$ Adaption | 0.139 | 0.923 | 0.887 | 0.517 | 0.144 | 0.955 | 0.830 | 0.420 |
| $w/o$ Symbolic | 0.135 | 0.933 | 0.898 | 0.536 | 0.140 | 0.959 | 0.843 | 0.414 |
| $w/o$ First Translation Model | 0.126 | 0.932 | 0.894 | 0.534 | 0.146 | 0.948 | 0.836 | 0.417 |

Table 12: Automatic evaluation results for additional ablation on the Original RobotHow and WikiHow. Metrics are computed between the annotated programs and the predictions.

**Motivation of Evaluation Metrics** Since the nature of the procedural planning task can be open-domain in that the golden plans may not be unique. This leads to the challenge that common automatic metrics proposed in natural language task are not perfect to evaluate procedural planning. The same observations of such challenge to directly judge the system using automatic metrics are discussed in LLMaP(Huang et al., 2022) as well. We assume that the human evaluation on `Coverage` and `Order` can reflect how well the procedural plans are close to human annotated program, because the human annotators are required to determine whether the task can be completed in any reasonable scenario using the procedural plans explicitly. Thus we provide both the automatic evaluation and human evaluation on two aspects `Coverage` and `Order`, with description in the **Metrics** paragraph in Section 4.1.

**Evaluation on Success Rate Metric** To make human evaluations more intuitive, we provide an additional `Success Rate` metric to show whether the procedural plans can successfully implement the task, which focus more on the success rate instead of the coverage or the order of the plans. We show the `Success Rate` evaluations on the baselines and our method in Table 11. The assignment layout template for workers is shown in Figure 8.

**More Ablation** To verify the contribution of the first translation language model $LM_T$ that translates the knowledge prompt $P_G$ into admissible one $\hat{P}_G$, we conduct an additional ablation experiment by simply removing the first $LM_T$ and replacing $\hat{P}_G$ with $P_G$ to prompt the LLM for procedural planning. We provide results with comparisons to other ablations in Table 12.

**Results on Counterfactual Task Samples** We show automatic evaluation results on counterfactual RobotHow in Table 13.

| Model | Initial Configuration | | | | Intermediate Step | | | | Final Goal | | | |
| --- | --- | --- | --- | --- | --- | --- | --- | --- | --- | --- | --- | --- |
| | S-BLEU | WMD | BERT-f1 | ROUGE-f1 | S-BLEU | WMD | BERT-f1 | ROUGE-f1 | S-BLEU | WMD | BERT-f1 | ROUGE-f1 |
| Chain (Wei et al., 2022) | 0.125 | 0.906 | 0.875 | 0.518 | 0.136 | 0.926 | 0.892 | 0.550 | 0.063 | 0.918 | 0.857 | 0.467 |
| LLMaP (Huang et al., 2022) | 0.148 | 0.929 | 0.887 | 0.566 | 0.141 | 0.886 | 0.902 | 0.547 | 0.070 | 0.928 | 0.868 | 0.490 |
| PLAN (Ours) | **0.169** | **0.934** | **0.897** | **0.570** | **0.183** | **0.953** | **0.913** | **0.590** | **0.082** | **0.934** | **0.873** | **0.493** |

Table 13: Automatic evaluation results on the Counterfactual RobotHow with language model GPT2.

# D    Qualitative Examples

## D.1    Intermediate Output

We provide running examples with intermediate output for each module in the following paragraph. First, we show the intermediate output of input task $T$, the subgraph $G_s$ depicted in the tuple of the start node, relation type, tail node and edge weight, the knowledge prompt $P_G$ and the translated one $\hat{P}_G$ as below:

- Input task $T$: Take shower.

- Human-annotated Plan Reference: Step 1: Walk to bathroom. Step 2: Walk to clothes dress. Step 3: Find clothes dress. Step 4: Put off clothes dress. Step 5: Find shower. Step 6: Enter shower. Step 7: Find soap. Step 8: Grab soap. Step 9: Scrub soap. Step 10: Put back soap. Step 11: Leave shower. Step 12: Find towel. Step 13: Grab towel. Step 14: Wipe towel. Step 15: Find clothes dress. Step 16: Put on clothes dress.

- Task-relevant subgraph $G_s(N_{head}, R_e, N_{tail}, E_w)$: (take a shower, HasLastSubevent, dry off, 6.0); (bathe, HasLastSubevent, dry off, 6.0); (take a shower, HasPrerequisite, take out your clothes, 4.47); (take a shower, HasSubevent, get clean, 4.47); (take a shower, HasPrerequisite, take your clothes off, 3.46); (go to a party, HasPrerequisite, take a shower, 2.82); (play lacrosse, HasLastSubevent, take a shower, 2.82); (get clean, HasPrerequisite, take a shower, 2.82); (take a shower, MotivatedByGoal, wash your hair, 2.82); (play sports, HasLastSubevent, take a shower, 2.82); (go to the hairdresser, HasPrerequisite, take a shower, 2.82); (take a shower, HasPrerequisite, turn on the water, 2.0); (have a bath, HasLastSubevent, dry off, 2.0); (get wet, HasSubevent, dry off, 2.0); (become more clean, HasLastSubevent, dry off, 2.0); (take a shower, HasSubevent, wash your hair, 2.0); (take a shower, HasLastSubevent, turn off the water, 2.0); (become more clean, HasLastSubevent, dry off, 2.0); take a shower, HasLastSubevent, put your clothes on, 1.0); (take a shower, HasSubevent, use shampoo, 1.0); (take a shower, HasSubevent, wash behind your ears, 1.0); (take a shower, HasSubevent, wash your body, 1.0); (take a shower, HasPrerequisite, go to the bathroom, 1.0); (take a shower, HasPrerequisite, go to the bathroom and undress, 1.0); (take a shower, HasPrerequisite, step into the shower, 1.0); (take a shower, HasPrerequisite, soap up, 1.0); (take a shower, HasPrerequisite, bring some soap, 1.0); (take a shower, HasPrerequisite, bring some shampoo, 1.0); (take a shower, HasLastSubevent, towel yourself off, 1.0); (take a shower, HasPrerequisite, remember to take your towel, 1.0);

- Knowledge prompt $P_G$: Step: take out your clothes. Step: set clean clothes. Step: go to the bathroom. Step: go to the bathroom and undress. Step: take your clothes off. Step: turn on the water. Step: bring some soap. Step: bring some shampoo. Step: remember to take your towel. Step: get clean. Step: wash your hair. Step: use shampoo. Step: wash behind your ears. Step: wash your body. Step: turn off the water. Step: dry off. Step: become more clean. Step: put your clothes on.

- Translated knowledge prompt $\hat{P}_G$: Step: find clothes underwear. Step: wash shower. Step: walk to bathroom. Step: look at shower. Step: take off clothes underwear. Step: wash shower. Step: find soap. Step: grab soap. Step: scrub shower. Step: find shampoo. Step: grab shampoo. Step: find towel. Step: wipe shower. Step: wash hair. Step: wipe shampoo. Step: scrub shower. Step: wash body. Step: switch off washing machine. Step: scrub shower. Step: wipe shower. Step: put on clothes underwear.

- Generated plan $S_T$: Step 1: Find clothes underwear. Step 2: Walk to Bathroom. Step 3: Take off clothes shirt. Step 4: Scrub shower. Step 5: Pour shampoo into hair. Step 6: Wipe shampoo. Step 7: Wipe hair. Step 8: Wash body. Step 9: Find Towel. Step 10: Put on clothes underwear.

## D.2    Predicted Procedural Plans

More qualitative examples of final predicted procedural plans that are randomly selected are provided. Table 14 show random samples on the original dataset. Table 15 show random samples on the counterfactual datasets with the Intermediate Step intervention method. And Table 16 shows random

samples on the counterfactual RobotHow with the Initial Configuration and Final Goal intervention methods.

| Model | Program |
|---|---|
| | **RobotHow** Task: Play Games |
| Human | Step 1: Walk to bedroom.Step 2: Walk to video game console.
Step 3: Find video game console.Step 4: Walk to video game console.
Step 5: Switch on video game console.Step 6: Find television.
Step 7: Switch on television.Step 8: Find video game controller.
Step 9: Grab video game controller.Step 10: Find chair.
Step 11: Walk to chair.Step 12: Sit on chair |
| Chain | Step 1: Put chef knife on water glass.Step 2: Find sink. |
| LLMaP | Step 1: Walk to bedroom.Step 2: Walk to video game console.
Step 3: Find video game console |
| PLAN | Step 1: Walk to video game controller.Step 2: Find video game controller.
Step 3: Switch on video game controller.Step 4: Find tv.Step 5: Switch on tv. |
| | **WikiHow** Task: How to Be an Organized Artist |
| Human | Step 1: Keep related supplies in the same area.,.
Step 2: Make an effort to clean a dedicated workspace after every session.,.
Step 3: Place loose supplies in large, clearly visible containers.,.
Step 4: Use clotheslines and clips to hang sketches, photos, and reference material.,.
Step 5: Use every inch of the room for storage, especially vertical space.,.
Step 6: Use chalkboard paint to make space for drafting ideas right on the walls.,.
Step 7: Purchase a label maker to make your organization strategy semi-permanent.,.
Step 8: Make a habit of throwing out old, excess, or useless stuff each month. |
| Chain | Step 1: Feel for a dense, kinky coat.
Step 2: Look for rounded ears with crimped inner hairs. |
| LLMaP | Step 1: Make sure that all money has been covered on both end.
Step 2: Check your bill and banking statements regularly.to ensure that your bills are
getting paid as they should be, you should check all statements that you receive.
Step 3: Keep track of your expenses.Step 4: Keep track of your income. |
| PLAN | Step 1: Prioritize your goals for the workshop.Step 2: Prepare to create your neopoprealist mural.
Step 3: Organize your computer-based materials.Step 4: Have a clear plan.
Step 5: Work on being the best you.
Step 6: Paint patterns with brushes of suitable size for the particular portion of work you are painting.
Step 7: Put together a schedule and chart.
Step 8: Buy your investment art when you feel confident of its worth, its price and its ability to grow
in value.
Step 9: Make a clean brea.
Step 10: First, you will need to make sure you have all of the materials listed below. |
| | **WikiHow** Task: How to Use Saddle Soap |
| Human | Step 1: Read the manufacturer's directions., Step 2: Avoid lye heavy products.,
Step 3: Make your own saddle soap. |
| Chain | Step 1: Give her a safe space.Step 2: Let him exercise.
Step 3: Be kind to him.Step 4: Offer him a treat. |
| LLMaP | Step 1: Dip a damp cloth in the saddle soap.
Step 2: Dampen several cloths and work them into a lather with the saddle soap.
Step 3: Dip a cloth into the mixture.Step 4: Rub the product on with a cloth.
Step 5: Clean the grime off with the cloth.Step 6: Dampen a cloth with water.
Step 7: Rub a damp microfiber cloth over the surface.Step 8: Use a damp cloth to wipe once again.
Step 9: Dampen a towel with lukewarm water.Step 10: Wipe the area clean with a damp towel.
Step 11: Soak a towel in warm water. |
| PLAN | Step 1: Brush your horse.Step 2: Clean the saddle.
Step 3: Start by using the first cloth to rub down the leather, making sure you remove any built
up grass and dirt.
Step 4: Rub the leather down so you remove most of the oil and wax.
Step 5: Wipe off excess oil.Step 6: Rinse the leather.
Step 7: Clean any other leather tack items (martingale, breastplate, draw reins, cruppers, etc. |

Table 14: Showcases of procedural steps predicted by different models with GPT2 as the base LLM.

| Model | Program |
|---|---|
| | **RobotHow** Intermediate Step
Turking (Find computer.). |
| Human | Step 1: Walk to home office.
Step 2: Walk to desk.
Step 3: Find chair.
Step 4: Sit on chair.
Step 5: Find computer.
Step 6: Switch on computer |
| Chain | Step 1: Switch on computer.
Step 2: Switch off computer. |
| LLMaP | Step 1: find computer. |
| PLAN | Step 1: Walk to chair.
Step 2: Turn to computer.
Step 3: Look at computer.
Step 4: Switch off computer.
Step 5: Switch on computer. |
| | **WikiHow** Intermediate Step
Task: How to Become an Art Investor (Figure out what you are willing to pay for art, before going to an auction house.). |
| Human | Step 1: Start with some experience or interest in art.,.
Step 2: Understand the difference between art collectors, art investors and art speculators.,.
Step 3: Figure out what you are willing to pay for art, before going to an auction house.,.
Step 4: Pay attention to what schools of art are selling well, and which are down.,.
Step 5: Focus art investments on fine art paintings, rather than decorative art.,.
Step 6: Reach out to trusted auction houses and dealers when you are looking to buy art.,.
Step 7: Buy your investment art when you feel confident of its worth, its price and its ability to grow in value.,.
Step 8: Study how art is properly stored.,.
Step 9: Have your art investments appraised occasionally.,.
Step 10: Consider renting out your art investments.,.
Step 11: Understand that selling an art investment can take time. |
| Chain | Step 1: See art being sold directly by artists by clicking on the "direct from the artist" link.
Step 2: Look at the price.
Step 3: Look at the writer's website.
Step 4: Create a portfolio of your work.
Step 5: Look at price again. |
| LLMaP | Step 1: Create a sketch in the neopoprealist manner of the future mural on a small piece of paper 8"x10" using the black ink pen.
Step 2: Prepare to create your neopoprealist mural.
Step 3: Prepare your paint.
Step 4: Begin your project with a design.
Step 5: Produce a scaled down version of your finished mural.
Step 6: Prepare the wall to be painted.
Step 7: Allow the background and base coats to dry.
Step 8: Draw the lines, then fill the appeared section with different repetitive patterns (examine the images above).
Step 9: Paint patterns with brushes of suitable size for the particular portion of work you are painting.
Step 10: Clean up the lines and shapes as needed.
Step 11: Seal the mural if needed.
Step 12: Be inspired and it will help you succee.
Step 13: This is where you get to be the most creative. |
| PLAN | Step 1: Figure out what you are willing to pay for art, before going to an auction house.
Step 2: Sell your art online.
Step 3: Go to the auction house.
Step 4: Buy it.
Step 5: Install the first floor.,repeat
steps 1-4 to add more floor.
Step 6: Enter in the second round values.
Step 7: Get the 3-carat diamond.
Step 8: Know the four cs of diamonds.
Step 9: Find a reputable diamond appraiser in your area. |

Table 15: Showcases of procedural steps predicted by different models with GPT2 as the base LLM under counterfactual setting (Intermediate Step).

| Model | Program |
|---|---|
| | **RobotHow** Initial Configuration
Task: Play games in dining room. |
| Human | Step 1: Walk to dining room.Step 2: Walk to video game console.
Step 3: Find video game console.Step 4: Walk to video game console.
Step 5: Switch on video game console.Step 6: Find television.
Step 7: Switch on television.Step 8: Find video game controller.
Step 9: Grab video game controller.Step 10: Find chair.
Step 11: Walk to chair.Step 12: Sit on chairTask: Play games in dining room.
Step 1: Walk to video game controller.Step 2: Put video game controller on diningtable.
Step 3: Put boardgame on kitchen table.Step 4: Put boardgame on diningtable. |
| Chain | Step 1: Walk to dining room.Step 2: Walk to sauce pan.
Step 3: Pour bottle water into dish bowl.Step 4: Walk to water.
Step 5: Walk to carrot.Step 6: Walk to food salt. |
| LLMaP | Step 1: Walk to kitchen.Step 2: Walk to food food.Step 3: Walk to dining room. |
| PLAN | Step 1: walk to dining room.Step 2: find diningtable.Step 3: walk to board game.
Step 4: put video game controller on diningtable.Step 5: put food snack on diningtable.
Step 6: put board game on diningtable.
Step 7: put boardgame on diningtable.Step 8: put board game on kitchen table.
Step 9: find video game console.Step 10: put glass on board game.
Step 11: grab video game controller.Step 12: put glass on boardgame. |
| | **RobotHow** Final Goal
Task: Turn light off and clean |
| Human | Step 1: Walk to bedroom.Step 2: Walk to light.Step 3: Switch off light.
Step 1: Walk to home office.Step 2: Walk to rag.Step 3: Find rag.
Step 4: Grab rag.Step 5: Walk to desk.Step 6: Find computer.
Step 7: Wipe computer.Step 8: Wipe desk.Step 9: Put back rag. |
| Chain | Step 1: Walk to kitchen.Step 2: Walk to cooking pot.Step 3: Walk to water.
Step 4: Walk to dishwasher. |
| LLMaP | Step 1: Put light bulb on bowl.Step 2: Switch off light bulb.Step 3: Switch on light. |
| PLAN | Step 1: plug out lighting.Step 2: put cleaning solution on desk.
Step 3: find dish soap.Step 4: scrub light switch.
Step 5: wipe lighting. |

Table 16: Showcases of procedural steps predicted by different models with GPT2 as the base LLM under counterfactual setting (Initial Configuration, Final Goal).

# E    DISCUSSION

## E.1    LIMITATIONS

Though pointing out a direction to prompt out actionable knowledge in large-scale pre-trained language models with external commonsense knowledge, the limitations of reasoning long-horizon procedural plan still exist. Existing datasets for procedural planning like WikiHow and RobotHow are all monolingual supporting only English goals and plans. In the future, it is important to expand these datasets or having novel datasets that support multiple languages used across the world. The inherent difference between these languages may also result in different planning strategies in granularity or abstraction levels, which is potentially challenging. In addition, the long-horizon and complex composite tasks still remain challenging for the existing procedural planners.

Above limitations are discussed mainly based on the challenges of procedural planning task. In addition, there are limitations of our implementation that are guided by our causal analysis. First, the coverage of the leveraged external resources is limited, which is common in a knowledge-enhanced system. This may result in the wrong understanding of the task and produce not reasonable procedural plans. For example, the knowledge of the word "Turking", which refers to "The act or process of performing small tasks using the Amazon Mechanical Turk service." according to Wiktionary, is not covered in the external resources (e.g., ConceptNet). Since our proposed system does not assume specific external resources. It is plausible in the future if we utilize more powerful external resources (e.g., Wiktionary). Second, the hop number and the threshold of the multi-hop retrieval in task-relevant subgraph sampling is currently a configured hyperparameter. This may result in not ideally constructed prompt. The future work could instead make these hyperparameters learnable on each task domain, and also explore the pros and cons between end-to-end commonsense-infused prompt versus neuro-symbolic constructed prompt.

## E.2    FAILURE ANALYSIS

We discuss detailed failure modes and examples with analyses below. For example, the predicted procedural plan on task "Turking", which refers to "The act or process of performing small tasks using the Amazon Mechanical Turk service." according to Wiktionary. We compare the predicted procedural plan on this task among baselines and our method: (1) The ground truth plan is "Task: Turking. Step 1: Walk to home office.Step 2: Walk to desk.Step 3: Find chair.Step 4: Sit on chair.Step 5: Find computer.Step 6: Switch on computer" (2) The plan predicted by Chain baseline is empty. (3) The plan predicted by LLMaP baseline is "Task: Turking. Step 1: Put teddybear on oven." (4) Our prediction is "Task: Turking. Step 1: Eat food turkey. Step 2: Drink water. Step 3: Sleep." We can see that for the "out-of-knowledge" task, our method also lead failure planning. We assume this is mainly due to the limited knowledge in external resources, as discussed in the Appendix E.1, and this main failure mode can be avoided by introducing larger external resources (e.g, Wiktionary), similar as other knowledge-enriched methods.

## E.3    ETHICAL CONSIDERATIONS

We hope to de-bias the procedural planning to avoid misleading either humans or robots with daily life instructions, which may result in unsafe situations. The cultural bias behind these datasets can be a critical issue for future work. As the ground truth planning steps usually reflect the culture shared by the English-speaking group, other cultures may have a completely different practical consideration that leads to different orders of these steps or even novel steps that are not proposed by the LLMs we utilized in this paper. In the future, we will consider cultural bias as a proxy variable so that we could adjust the implicit knowledge from LLM or commonsense from external sources according to the different needs of cultural backgrounds.

