# OpenReview forum: "Neuro-Symbolic Procedural Planning with Commonsense Prompting"
_ICLR.cc/2023/Conference — ICLR 2023 notable top 25%_

### Official Review · Reviewer_UGe5 · 2022-10-13

**Confidence:** 5
**Correctness:** 4
**Technical Novelty And Significance:** 3
**Empirical Novelty And Significance:** 3
**Recommendation:** 10

**Clarity, Quality, Novelty And Reproducibility:**

As we have discussed in the prior section, reproducibility is good.
In overall, the novelty should be good because introducing casual models and neural symbolic methods should still be novel.
The major limitation is the clarity and the paper presentation. I tend to the acceptance part considering the good directions and good ideas. However, if I do not see a major revision in the rebuttal phase in the writing parts clarifying the related concerns, I might decide to reject this paper.

**Strength And Weaknesses:**

## Strengths:
1. The procedural planning task is interesting and fundamental. Through procedural planning, we can understand how the neural networks understand the task structure and generate the corresponding plans. The counteract cases are also interesting and touch the core of AI.
2. The source code is provided, so this work might be reproducible.
3. The idea of using causal models to prevent relational biases and keeping order in LLMs is very important and is introduced with good examples.
4. The improvements in the experiments seem to be large. I like the illustration of the results introduced in Table 5. Sadly, some results are omitted in Table 5. On the contrary, I suggest the paper should have a large table showing the concrete procedural results to help understand each part of the model. Try always to be concrete and precise.

## Weaknesses:
I am confused by this paper's methodology a lot of times. In general, this paper could be challenged in the following positions:

1. Probably, the proposed model is not technically sound. In fact, many of the components are not well-defined, and the system looks too complicated. The overall model is not ablated well and is full of tricks.
2. The proposed model seems to borrow a lot of big concepts, such as neural-symbolic or casual models. However, the specific technical contributions are not very clear, and many problems remain unaddressed. In general, the system looks too complicated. This is not a good paper. A good paper should stick to one major point and show the advantage of this point over previous baselines.
3. Human evaluations are not very objective and cannot be easily reproduced. I fail to see a clear motivation for using expensive human evaluations. Instead, showing a lot of generated procedures is far more beneficial than showing a lot of numbers.

The specific comments are:

1. In the task definition, what are the actions, and what are the object sets? I think in real natural language, the set of actions (described in natural language) or objects can be extremely large. How can the formulation handle this? I did not see the specific usage of these variables. Probably the problem is over-formulated.
2.  The authors use the same variable D for task D_T and confounding variable D. This is hard to decode.
3.  I can hardly understand the implementation of the SCM. What are the input and output of the SCM? Is there any replacement (e.g. standard transformers) for this module? What is the sample output of the SCM? Is there any reference to the SCM? Take Figure 2 as an example; how does the SCM forward the reasoning path? How does the SCM encode the action and object? How does the SCM produce the output? These problems are still unclear to me.
4. Why does procedural planning need so many steps (five-stage pipelines)? I see that motivation is to fuse different motivations (task definition, previous steps, and external knowledge). If so, why not encode this information in parallel so that the fused (e.g. concatenation) information can be used in downstream tasks?
5. Algorithm 1 is a waste of space, and it is better to replace it with a figure (or make figure 3 better).
6. Many concrete designs, such as the symbolic executors, are missing. It seems that the symbolic executors are trivial and can be easily learned. In other words, I do not think the "neural-symbolic" stuff in the title plays an important role in this system. In fact, what is neural symbolic in this line of work? Why is it important? How to connect the big concept with the experimental results with the LLMs? These questions are far more important to show than the ad-hoc model designs.

**Summary Of The Paper:**

This paper presents a procedure planning model which enforces the logical order. The authors claim that they can parse the goal to subgraphs and translate the knowledge into admissible knowledge. The proposed model also learns casual relationships via an SCM for procedural planning. Experimental results show the proposed model surpasses the original baselines by a large margin.

**Summary Of The Review:**

Addressing an important task with promising novel technics. Writing needs to be improved, and more insights should be given.
-------------------------------------
After reading the responses, I think most of my concerns are addressed. Therefore I raised my score.

---

> ### Author Response · Authors · 2022-11-15
> **Response to Official Review of Paper822 by Reviewer UGe5 Part3**
>
> > **R5C4.**  About the implementation and definition of the SCM.
>
> We follow the definition of the Structural Causal Model (SCM) in causal inference literatures ([1,2]). The earliest version of SCM is introduced in [3]. The SCM is commonly represented with causal graphs (entension of directed acyclic graphs) that express the causal relationships between variables in a system ([4,5]). We include the causal prelimiaries in Appendix A.1.
> For better clarity, we refer readers to these definitions in the introduction section as "Please refer to Appendix A.1 for causal preliminaries, Appendix A.2 for the back-door adjustment definition, and Appendix A.3 for the front-door adjustment definition." in the revision main paper.
> Basically we first define the SCM in Figure 2 to describe the causal relationship among variables in the procedural planning task. The SCM is actually a causal graph modeling with abstract nodes and edges, instead of something like a neural network "model" (e.g., transformer) that take some input and produce the output. After defining our SCM, we propose to apply front-door adjustment to solve the spurious correlation (Figure 2(b)). To achieve the front-door adjustment, we devise an architecture (Figure 3) which can be separated in $5$ stages (Section 3.1 and 3.2).
>
> [1] Noah Weber, Rachel Rudinger, and Benjamin Van Durme. 2020. [Causal Inference of Script Knowledge](https://aclanthology.org/2020.emnlp-main.612). EMNLP, pages 7583–7596, Online. Association for Computational Linguistics.
> [2] X. Yang, H. Zhang and J. Cai, "Deconfounded Image Captioning: A Causal Retrospect," in _IEEE Transactions on Pattern Analysis and Machine Intelligence_, doi: 10.1109/TPAMI.2021.3121705.
> [3] [Wright, S. (1921). “Correlation and causation”. Journal of Agricultural Research. 20: 557–585.](https://naldc.nal.usda.gov/download/IND43966364/pdf)
> [4] [Structural Causal Models](https://www.causalflows.com/structural-causal-models/)
> [5] [Edward N. Zalta, 2018, "Causal Models", The Stanford Encyclopedia of Philosophy](https://plato.stanford.edu/entries/causal-models/)
>
>
> > **R5C5.** Why does procedural planning need so many steps (five-stage pipelines)?..why not encode this information in parallel so that the fused (e.g. concatenation) information can be used in downstream tasks?
>
> Please refer to the response at **R5W(eakness)1.** and **R5W2.**, where we explain why the complexity of the baselines and our approach are comparable. Basically, our implementation at Section 3 is guided by the causal analysis in Section 2. We separate the sysmte into a more fine-grained level ($5$ stages) to make it easier to connect with the intuition from Section 2, Figure 2 and Figure 4. Each stage either refer to the operation in the SCM or the generation process. We're fusing the  information togerther, but the motivation is more into how we inject the external knowledge to mitigate the biased pre-trained knowledge $D$.
>
> > **R5C6.**  Algorithm 1 is a waste of space, and it is better to replace it with a figure (or make figure 3 better).
>
> We added an additional zoomed-in algorithm block for Section 3.1 in Algorithm 1 to help parse the computation flow in Section 3 and the architecture in Figure 3.
>
> > **R5C7.**  Many concrete designs, such as the symbolic executors, are missing. ... In fact, what is neural symbolic in this line of work? Why is it important? How to connect the big concept with the experimental results with the LLMs? These questions are far more important to show than the ad-hoc model designs.
>
> We first show our causal analysis in Section 2 and then we show how we implement the operations in SCM in Section 3. Specifically, we leverage a neuro-symbolic approach to construct commonsense-infused prompts, which preserve structural knowledge. We show the ablation without a symbolic module in Table 3, which experiences the performance drop in the LLMs. Besides, we add more explained details of the implementation of our neuro-symbolic approach, including the symbolic executors in Section 3.1 Stage 2.
>
> ### Clarity, Quality, Novelty And Reproducibility
> > - "The major limitation is the clarity and the paper presentation." "However, if I do not see a major revision in the rebuttal phase in the writing parts clarifying the related concerns, I might decide to reject this paper."
>
> **We've improved the presentation of the paper by making a thorough revision as suggested.**
> We thank the reviewer for recognizing the good directions and good ideas of our work. And we've made a revision that addresses each reviewer's concerns point-wisely for clarity. In addition, we made a major revision of the writing and absorb writing improving the opinions of researchers from different backgrounds to make sure our paper is more clear and easy to follow.

---

> > ### Comment · Reviewer_UGe5 · 2022-11-15
> > **Potentially great paper; I am willing to see the revised paper.**
> >
> > Dear authors,
> > Thanks for your reply.  Most of the questions are answered adequately. In general, this paper and its directions have great potential, and I cannot wait to advertise this paper to my colleagues if it finally gets accepted. We will discuss others with the negative reviewers about other issues. More suggestions:
> > 1. Post a comment after you finish the paper revision. Other reviewers also mention about writing issues and we need to examine the revised version before making decisions.
> > 2. Give FULL examples in Table 3. Table 3 shows many "...", which does not fully reveal the performance difference between the methods.
> > Yours,
> > UGe5

---

> > > ### Author Response · Authors · 2022-11-15
> > > **Thank you for your recognition and suggestions for our work.**
> > >
> > > Dear Reviewer,
> > >
> > > Thank you for your interest in our work. We're willing to keep on addressing issues and concerns.
> > > Thank you for more suggestions:
> > > 1. Sure, we will post a comment after finishing the revision.
> > > 2. We will provide full examples in Table 3 as suggested.
> > >
> > > Best Regards,
> > > Authors of Paper822

---

> ### Author Response · Authors · 2022-11-15
> **Response to Official Review of Paper822 by Reviewer UGe5 Part2**
>
> > **R5W3.**  Human evaluations are not very objective and cannot be easily reproduced. I fail to see a clear motivation for using expensive human evaluations. Instead, showing a lot of generated procedures is far more beneficial than showing a lot of numbers.
>
> 1. Motivation for using human evalutions: Our motivation of using human evaluations share the same spirit with the motivation in the baseline Zero-shot Planner (LLMaP) paper (Huang, et al.): "Unlike most embodied environments where the completion of a task can be easily judged, the ambiguous and multimodal nature of natural language task specification makes it impractical to obtain a gold-standard measurement of correctness. Therefore, we conduct human evaluations for the main methods. ".
> We add a new paragraph that explain the motivation why we use human evaluations in the revision in Appendix C.2: "Since the nature of the procedural planning task can be open-domain in that the golden plans may not be unique. This leads to the challenge that common automatic metrics proposed in natural language task are not perfect to evaluate procedural planning. We assume that the human evaluation on *Coverage* and *Order* can reflect how well the procedural plans are close to human-annotated program, because the human annotators are required to determine whether the task can be completed in any reasonable scenario using the procedural plans explicitly. Thus we provide both the automatic evaluation and human evaluation on two aspects  with description in the Metrics paragraph in Section 4.1."
> 3. Generated procedures: In addition to showcases in Table 3, we provide a lot of generated procedures of baselines and our method based on two LLMs architectures on two datasets under standard and counterfactual setting in Table 10-14. We also extend Table 10 and Table 11 with more examples to help readers compare the qualitative results.
>
> ### Specific Comments
> > **R5C(omment)1.**  In the task definition, what are the actions, and what are the object sets? I think in real natural language, the set of actions (described in natural language) or objects can be extremely large. How can the formulation handle this? I did not see the specific usage of these variables. Probably the problem is over-formulated.
>
> Yes, the actions and objects set are extremely large and we do not specify that explicitly. Indeed, the system is aware of the admissible plans set $\bar{S}$ for each task domain. We modify the problem definition as "There exists certain admissible plans $\bar{S}$, which is a fixed set constrained by the task domain $M_T$ (e.g., the affordance of the interacted objects)." And we updated the description in Stage 5 paragraph in Section 3.2 to remove the over-formulated $A$ and $O$ for more accurate definition.
>
>
> > **R5C2.**  The authors use the same variable D for task D_T and confounding variable D. This is hard to decode.
>
> We change the notation for the task domain from $D_T$ to $M_T$, to seperate from the variable $D$ for better clarity.
>
> > **R5C3.** This paper is related to program-guided works [1-2]. A deep discussion is recommended.
>
> These two program-guided works aim to utlize human written programs in a formal language as a precise and expressive way to specify task, and thus avoid ambiguity of natural language instructions. Our method is guided by the intuition of our thorough causal analysis of the procedural planning task and come to the solution of constructing prompt from external resources in a neuro-symbolic approach.
> We add the discussion about these two program-guided workds in related work (Procedural Planning in Section 5) in the revision.

---

> ### Author Response · Authors · 2022-11-15
> **Response to Official Review of Paper822 by Reviewer UGe5 Part1**
>
> ### Weakness
> > **R5W(eakness)1.**  Probably, the proposed model is not technically sound. In fact, many of the components are not well-defined, and the system looks too complicated. The overall model is not ablated well and is full of tricks.
>
>  1. We re-write the Section 2 and 3 to make it more clear how each component function and contribute to the system.
>  2. We're showing the system at a fine-grained level, so that we could show the implementation details guided by the analysis of SCM. This fine-grained depiction of our system make it seems very complex, however this is comparable with other baselines. For example, the method in Zero-shot planner (Huang et al.) can be separted as $4$ stages: (1) retrieve the task-relevant program from held-out exemplars (2) Aggregate the prompt with previous steps as temporally-extended prompt for predicting next step (3) Semantic Generation (similar as Stage $4$ in our paper) (4) Admissible Step Translation (the same as Stage $5$ in our paper). The first two stages can be merged into one, however we separete it into two stages to more clearly explain their connection with the causal analysis in Section 2. Thus we claim that we did not introduce a lot complexity into the sysmte.
>  3. As illustrated above, the complexity of our method and the baselines are comparable. Notice that the baslines addtionally need to extract the program-related exemplars. What we do is to replace that part by constructing  knowledge-aware prompt from the external resources instead of retrieving the expensive exemplars required by the baselines.  And how we extract it is guided by the analysis of Structural Causal Model.
>  4. We also empircally show that the computing cost of the baselines and our method is comparable in Table 8 and Table 9.
>
> > **R5W2.**  The proposed model seems to borrow a lot of big concepts, such as neural-symbolic or casual models. However, the specific technical contributions are not very clear, and many problems remain unaddressed. In general, the system looks too complicated. This is not a good paper. A good paper should stick to one major point and show the advantage of this point over previous baselines.
>
> 1. We're showing the system at a fine-grained level, so that we could show the implementation details guided by the analysis of SCM. This fine-grained depiction of our system make it seems very complex, however this is comparable with other baselines. For example, the method in Zero-shot planner (Huang et al.) can be separted as $4$ stages: (1) retrieve the task-relevant program from held-out exemplars (2) Aggregate the prompt with previous steps as temporally-extended prompt for predicting next step (3) Semantic Generation (similar as Stage $4$ in our paper) (4) Admissible Step Translation (the same as Stage $5$ in our paper). The first two stages can be merged into one, however we separete it into two stages to more clearly explain their connection with the causal analysis in Section 2. Thus we claim that we did not introduce a lot complexity into the system. Please also refer to the response at  **R5W(eakness)1.**.
> 2. The computing cost is ignorable as empirically shown in Table 8 and Table 9, which show the computing complexity of our method and the baselines are comparable.
> 3. The major point of our paper is to enable the system with the ability to be aware of the cause-effect relations in procedures. We describe the causal analysis in Section 2 and explain why external knowledge matters in procedural planning. Then we come to how to implement with the guidance of the causal analysis in Section 3. We leverage neuro-symbolic techniques to implement the intervention operations in the SCM. In general, Section 3 is our approach guided by the analysis in Section 2.
> 4. As illustrated in above three points, we claim that the complexity of our system is comparable as the baselines. At the same time we achieve significant improvement even without additional exemplars.

---

### Official Review · Reviewer_i21n · 2022-10-25

**Confidence:** 4
**Correctness:** 3
**Technical Novelty And Significance:** 4
**Empirical Novelty And Significance:** 3
**Recommendation:** 6

**Clarity, Quality, Novelty And Reproducibility:**

clarity : poor. And this is a huge problem because the writing prevented me from judging the work clearly.

In the introduction there's this block of text that reads

"adjustment (Hu & Li, 2021; Weber et al., 2020; Yue et al., 2020) are not applicable in our SCM. Instead, we build a mediator and implement it as a commonsense-infused prompt. Through the mediator, we can identify causal effects among goals and steps by investigating the indirect effect from the goals, which is essentially the frontdoor adjustment in causality (Pearl, 2009)."

What is it even saying? I have zero clue. What is a mediator? What is a commonsense-infused prompt? What are these "indirect affect from the goals" mean? What is "essentially the frontdoor adjustment" mean?

These are highly technical terms that mean very little unless explicitly defined. The reader tends to look for easy metaphors and intuitions on why your approach should work, and why intuitively it should work well. This passage sounds intuitive, yet it uses words that nobody know what they mean (yet), and ended up being just gibberish.

This confusion continued for the rest of the paper, making it hard for me to judge if it is worthwhile.

A re-write of the intro section is warranted, with a concrete example explaining why the proposed approach should work well, without the jargons.

I highly recommend the authors ask people outside of their immediate project -- walk down the hallway a few offices and knock on some doors -- ask these people to read the paper and give feedbacks, and adjust the paper based on what was confusing.

quality : unclear / potentially good. The human-evaluation is clearly stated, and I can feel confident in saying "the approach is performing better than the baseline". However, I would also like to make an assessment on "is this approach general? or is it domain-specific and hacky?", this is hard to judge as the work seemed very complex with many moving parts (in figure 3 there's 5 stages), and the writing isn't clear.

novelty : unclear / potentially good. same reason as above. If this work is generalizable to different domains with very little tweaking, then it definitely has merits most prior works that brings symbolic reasonings into neural models heavily rely on a DSL, i.e. a domain SPECIFIC language, and isn't really generalizable.

**Strength And Weaknesses:**

strength : the proposed approach works. they conduct a user study where they ask crowd workers to rate which agent, one with symbolic reasoning and one without, performed better on a task, and the crowd workers preferred the agent with symbolic reasoning. this result is solid and shows evidence of the proposed approach.

weakness :

the proposed method is may not be entirely novel. people have been adding symbolic reasoning to neural models for awhile, and the finding has always been : "If we can successfully 'hack' the underlying DSL that represented the set of tasks, adding symbolic reasoning would perform well". For instance, these works tend to follow the steps of: 1) identify a set of tasks that would be easily represented with symbolic execution, and 2) devote significant engineering efforts to construct the DSL and a symbolic interpreter to help the neural/llm model make better inferences/plans.

this work would be of significant contribution if it can show that steps 1) and 2) can be avoided by using a generic external knowledge base (as shown in figure 3). however the writing is too confusing I cannot be sure if that is the case or not.

**Summary Of The Paper:**

This work shows that by adding a dash symbolic reasoning to a neural model, it shows better performance w.r.t consistency and generalization. It is unclear to me how much engineering efforts are required to add these symbolic reasonings. It appears their approach is general, using a common External Knowledge Base, but the writing is confusing and I cannot tease out the details.

**Summary Of The Review:**

Overall, the paper is difficult to read, and as a result I cannot judge it properly.

---

> ### Comment · Reviewer_UGe5 · 2022-11-07
> **Good reviews.**
>
> I really enjoyed reading your comments. Thanks a lot!

---

> ### Author Response · Authors · 2022-11-15
> **Response to Official Review of Paper822 by Reviewer i21n**
>
> ### Weakness
> > **R4W(eakness)1.** the proposed method is may not be entirely novel. people have been adding symbolic reasoning to neural models for awhile, and the finding has always been : "If we can successfully 'hack' the underlying DSL that represented the set of tasks, adding symbolic reasoning would perform well". For instance, these works tend to follow the steps of: 1) identify a set of tasks that would be easily represented with symbolic execution, and 2) devote significant engineering efforts to construct the DSL and a symbolic interpreter to help the neural/llm model make better inferences/plans.
> > this work would be of significant contribution if it can show that steps 1) and 2) can be avoided by using a generic external knowledge base (as shown in figure 3). however the writing is too confusing I cannot be sure if that is the case or not.
>
> 1. The novelty and also the main point of our work is to conduct the first thorough causal analysis that show the spurious correlations among goals and steps in the procedural planning task. And previous methods that simply prompt the LLMs with the ememplars are not aware of the cause-effet relations in procedure. After defining the problem and SCM of procedural planning in Section 2, we propose a neuro-symbolic approach to implement the interventions toward the SCM to elicit the unbiased procedural knowledge from LLMs.
> 2. The symbolic module is utilized to preserve the structure that is benefitial for procedural planning. We're not manully providing any ground truth symbolic procedures to directly represent the task. And we're automatically generating the commonsense-infused prompts to replace with exemplars that really require manual efforts in the baselines.
> 3. We do not specify the domain of the tasks or knowledge bases. We're using a generic commonsense knowledge base, that do not direclty relate to the tasks in RobotHow or WikiHow but shared the spirit that these knowledge are required in human daily activities.
>
>
> ### Clarity, Quality, Novelty And Reproducibility
> > **R4C(larity)1.** Clarity
>
> We re-wrote the introduction section thoroughly and give a brief introduction to the terminologies, including what is a mediator in causality, how the commonsense-infused prompt is constructed to serve as such mediator, why the effects from the goals are indirect, and what is the front-door adjustment. We also point the readers to the appendix for more background in causality as "Please refer to Appendix A.1 for causal preliminaries (including explanation for SCM, confounder, mediator, spurious correlations), and Appendix A.3 for the front-door adjustment definition."
>
> In addition, we explain the decision behind each design in the introduction with some concrete examples to show why our approach perform well.
>
> We have made some major revisions to address the clarity and improve the writing of paper as suggested in the rebuttal revision. We also ask people from different backgrounds to give comments and help us revise on the presentation to adjust confusing part in the paper.
>
> > **R4C2.** Quality
>
> Yes, please also refer to **R4W(eakness)1.**. Our approach do not assume a specific domain and is a general approach for procedural planning.
>
> > **R4C31.** Novelty
>
> Our main contribution is we develop the first causal framework for procedural planning with thorough causal analysis. And to implement such causal framework (Section 2), we devise a neuro-symbolic approach to construct commonsense-infused prompts and control the generation of LLMs (Section 3) in procedural planning with awareness of cause-effect among goals and steps.

---

> > ### Comment · Reviewer_i21n · 2022-11-21
> > **making forward reference to appendix in the intro section isn't helpful for clarity**
> >
> > The purpose of the intro section is to have a self-contained, intuitive explanation of your work. You can't expect the first-time reader to skip ahead and read the appendix for crucial definitions that _needs to be understood_ to proceed. The appendix is for extra details that can be omitted entirely and the paper can still stands, reserved for those who are keen on replicating your work faithfully. So I'm reserving my scores on clarity as "poor".
> >
> > Since the authors claim their approach is domain agnostic and does not require specific engineering efforts, and the single knowledge base can serve multiple different domains without much hassle, I'm going to raise my score accordingly.
> >
> > In the end, I would still highly recommend a re-write of the introduction if the paper wants to reach a wider range of audiences and become a go-to reference for neuro-symbolic guided text generations.

---

> > > ### Author Response · Authors · 2022-11-22
> > > **Response to follow-up concerns of the clarity.**
> > >
> > > We agree that the intro should be self-contained. Thanks to the clarity suggesting in the comments, we made an effort to give a brief introduction to each term as suggested in the revision of introduction. Though we leave the reference to Appendix for the detailed definitions for terms in causality, the intro can still be understood with our brief and simplified explanation. We will keep on polishing the introduction as well as the other sections to ensure better clarity of paper.
> > >
> > > We thank the reviewer for recognizing our approach and raise the score accordingly.
> > >
> > > Thank you for the helpful suggestion. We will definitely improve the writing more to reach a wider range of audiences as suggested.

---

### Official Review · Reviewer_ai1E · 2022-10-30

**Confidence:** 2
**Correctness:** 3
**Technical Novelty And Significance:** 4
**Empirical Novelty And Significance:** 4
**Recommendation:** 8

**Clarity, Quality, Novelty And Reproducibility:**

- Clarity: some descriptions in the paper are currently not clear enough.
- Quality: the quality of the paper is good.
- Novelty: the contributions of the paper are novel.
- Reproducibility: Some additional details are required for one to reproduce the results in the paper. Partial code was provided but not the executable full code.


**Strength And Weaknesses:**

Strength:
1. Solid and reasonable idea: Based on the observation that due to potentially biased pre-training data, pre-trained knowledge in LLMs may confound the model to make wrong decisions when asking the model to generate a procedural plan of a task, the authors proposed to apply the frontdoor adjustment in causality to build a mediator and implemented it as a commonsense-infused prompt. The prompt obtained from their neuro-symbolic-based method allows the LLM to attend to the causal entities instead of the highly correlated ones for the next step generation.
2. Strong performance: the proposed method PLAN outperformed two recent SOTA methods LLMaP (“Language models as zero- shot planners”) and Chain (“Chain of thought prompting”) statistically significantly.
3. Quite thorough experimental study and analysis: for the experiments, the authors utilized several evaluation methods including human evaluations, two datasets, and several pre-trained LLMs.
4. The paper is well written and organized.

Weaknesses:
1. Discussion on the failure cases is currently missing. In addition, the generated procedural plan of the proposed method was shown, but it would be interesting and useful for readers to see the exact intermediate outputs of the proposed framework given an actual task from the evaluation dataset, e.g., $G_s$, $P_G$, $\hat{P}_G$, etc. In this way, readers may have a better understanding of the current capabilities of each module of the proposed framework.

2. Some minor issues, e.g.:
(1) “Note that in the path T →D→Si ←Pi, Si is a collider…” (second line of the “Stage1” paragraph), should it be “T←D”?
(2) It would be good to refer readers to the appendix for definitions of backdoor path, frontdoor adjustment, etc.
(3) How is the Symbolic Rule Set $R$ obtained?
(4) One more ablation experiment: what if removing the first Translation $LM_T$ and replacing $\hat{P}_G$ with $P_G$?
(5) Suggesting one relevant work: https://arxiv.org/abs/2205.11916 (manual exemplars not required).



**Summary Of The Paper:**

Summary:
- Existing large language models (LLMs)  require manual exemplars to acquire procedural planning knowledge in the zero-shot setting.
- The paper proposed a neuro-symbolic procedural PLANner (PLAN) with commonsense-infused prompts elicited from an external knowledge base (i.e., ConceptNet) to solve the pure-language-based procedural planning problem in a zero-shot manner.
- Human and automatic evaluations on WikiHow and RobotHow show the superiority of the proposed PLAN over the prior methods on procedural planning.


**Summary Of The Review:**

The paper proposed to generate commonsense-infused prompts elicited from an external knowledge base (i.e., ConceptNet) to allow LLMs to solve the pure-language-based procedural planning problem in a zero-shot manner. The key idea of this paper is novel and solid. The proposed method has a strong performance.

---

> ### Author Response · Authors · 2022-11-15
> **Response to Official Review of Paper822 by Reviewer ai1E**
>
> ### Weaknesses
> > **R3W(eakness)1.** Discussion on the failure cases is currently missing. In addition, the generated procedural plan of the proposed method was shown, but it would be interesting and useful for readers to see the exact intermediate outputs of the proposed framework given an actual task from the evaluation dataset, e.g.,  Gs,  PG,  P^G, etc. In this way, readers may have a better understanding of the current capabilities of each module of the proposed framework.
>
> 1. **Failure Cases**
> We extend Appendix E.1 for deeper discussion about the limitations and add a new Appendix E.2 for filure cases discussion. Please also refer to the response at **R1Q3.** and **R2W2RQ1** for the details, where we analyze the failure case of the task "Turking". The word "Turking" refers to "The act or process of performing small tasks using the Amazon Mechanical Turk service." according to Wiktionary.
> We recall the example shown in **R2W2RQ1.** and Appendix E.2 below:
> We compare the predicted procedural plan on this task among baselines and our method: (1) The ground truth plan is "Task: Turking. Step 1: Walk to home office.Step 2: Walk to desk.Step 3: Find chair.Step 4: Sit on chair.Step 5: Find computer.Step 6: Switch on computer" (2) The plan predicted by Chain baseline is empty. (3) The plan predicted by LLMaP baseline is "Task: Turking. Step 1: Put teddybear on oven." (4) Our prediction is "Task: Turking. Step 1: Eat food turkey.  Step 2: Drink water.  Step 3: Sleep." We can see that for the "out-of-knowledge" task, our method also lead failure planning.
> 3.  As suggested, We show exact intermediate outputs of the proposed framework given an actual task from the evaluation dataset in Appendix D.2.
>
> ### Some minor issues
> > **R3M(inor)I(ssues)1.** “Note that in the path T →D→Si ←Pi, Si is a collider…” (second line of the “Stage1” paragraph), should it be “T←D”?
>
> Yes, it's a typo. It should be "T←D". We fix it as "T←D→Si ←Pi" in the revision.
>
> > **R3MI2.** It would be good to refer readers to the appendix for definitions of backdoor path, frontdoor adjustment, etc.
>
> We thank the reviewer for this suggestion. We add more references for readers to the term definitions whenever it's needed in the revision. For example, we point readers to the Appendix as "Please refer to Appendix A.1 for causal preliminaries, Appendix A.2 for the back-door adjustment definition, and Appendix A.3 for the front-door adjustment definition." in the introduction section.
>
> > **R3MI3.** How is the Symbolic Rule Set  R  obtained?
>
> The Symbolic Rule Set $R$ is obtained by mapping the description of the relations (e.g., $AtLocation$ represent `A is a typical location for B, or A is the inherent location of B. Some instances of this would be considered meronyms in WordNet.') in external knowledge graph (e.g., ConceptNet) to symbolic operations (e.g., $Op\_AtLocation$). We add above details in Stage 2 paragraph in Section 3.1 in the revision.
>
> > **R3MI4.** One more ablation experiment: what if removing the first Translation  LMT  and replacing  P^G  with  PG?
>
> As suggested, to verify the contribution of the first translation language model $LM_T$ that translate the knowledge prompt $P_G$ into admissible one $\hat{P_G}$, we conduct an additional ablation experiment by simply remove the first $LM_T$ and replace $\hat{P_G}$ with $P_G$ to prompt the LLM for procedural planning. We provide results with comparison with other ablations in Table 13 in Appendix C.2.
>
> > **R3MI5.** Suggesting one relevant work:  [https://arxiv.org/abs/2205.11916](https://arxiv.org/abs/2205.11916)  (manual exemplars not required).
>
> This work shows promsing results on leveraging enormous zero-shot knowledge inside LLMs but seems only work on GPT3 architecture. Besides the generated plans are not structured in a desired format.
> We consider it as an interesting relevant work and cite this work in the **Prompt for Large Language Model** paragraph in Section 5 Related Works.
>
> ### Clarity, Quality, Novelty And Reproducibility
> For clarity, we've updated the draft thoroughly and addressing the comments point-by-point. For reproducibility, we will provide the detailed configuration files and executation script to help reproduce the results.

---

> > ### Comment · Reviewer_ai1E · 2022-11-20
> > **Thanks for the response. One explanation is still unclear to me.**
> >
> > "The Symbolic Rule Set is obtained by mapping the description of the relations to symbolic operations", how did you do the mapping exactly? And, "the Symbolic Executor acquires the neural information of each natural language node", why would the Symbolic Executor need *neural* information of node (I'm assuming "neural information" means vectorized representation of the natural language information of a node from a pre-trained language model)?
> >
> > I thought the process of constructing procedural prompt P_G by verbalizing the re-weighted G_s using R is basically: (1) list out the (head, relation, tail) triplets in G_s in the descending order of the adapted relation edge weights (that are greater than the threshold), (2) map each relation edge to a symbolic operation, given a *pre-defined* rule of how to map each relation type to a symbolic operation, (3) starting from the first triplet to the last after ranking and thresholding, given the symbolic operation of each relation edge which defines how to verbalize a triplet (or how to recursively navigate the graph if needed), verbalize the triplet into a natural language description. Am I accurate ? In what I described, I don't see which parts need the *neural* information of node.

---

> > > ### Comment · Reviewer_UGe5 · 2022-11-21
> > > **Good question.**
> > >
> > > It's a good question. I hope the authors can clarify this.
> > > Yours,
> > > UGe5

---

> > > > ### Author Response · Authors · 2022-11-22
> > > > **Response Update.**
> > > >
> > > > Thank you for pointing it out. We agree it's important to clarify this. We address this in the comments. And will update this in the final version of the paper.

---

> > > > > ### Comment · Reviewer_UGe5 · 2022-11-22
> > > > > **Makes sense.**
> > > > >
> > > > > I think the original draft is confusing at this point, and the authors' response is making more sense. I wish the authors to change the confusing terms and revise their draft before the camera-ready deadline.

---

> > > > > > ### Author Response · Authors · 2022-11-22
> > > > > > **Thank you.**
> > > > > >
> > > > > > Thank you! And we will surely continue revising the draft and improve writing before the camera-ready deadline.

---

> > > > > > > ### Comment · Reviewer_UGe5 · 2022-11-22
> > > > > > > **That's good.**
> > > > > > >
> > > > > > > I have increased my score from six to eight. Congrats!

---

> > > > > > > > ### Author Response · Authors · 2022-11-22
> > > > > > > > **Thank you for reviewing and supporting our paper!**
> > > > > > > >
> > > > > > > > Thank you for continually providing suggestions and actively participating in the discussion to help us revising the paper. Many thanks to your time and effort!

---

> > > ### Author Response · Authors · 2022-11-22
> > > **Thanks for pointing it out. We address it in the below comment and will update in the paper.**
> > >
> > > We thank the reviewer for pointing it out. We will address this concern as below:
> > >
> > > 1. Mapping the description of the relations to symbolic operations
> > > In ConceptNet 5, they define a set of relations that can aply to text in any langauge here: [Relations in ConceptNet5](https://github.com/commonsense/conceptnet5/wiki/Relations). The relations that we used to map as the symbolic operations include AtLocation, CapableOf, Causes, CausesDesire, HasPrerequisite, HasSubevent, and UsedFor, etc. We filter out the relations (/r/DistinctFrom, /r/DerivedFrom, /r/SymbolOf, /r/EtymologicallyRelatedTo, /r/EtymologicallyDerivedFrom) that are related to the linguistic instead of the commonsense knowledge required in household daily activities. We include these relations choice details in Stage1 Section 3.1 and **External Knowledge Base** in Appendix B.1.
> > > Based on these chosen relations, we can map the description of the relations to symbolic operations.
> > > Specifically, instead of manually defining some domain specific language of functions and relations, we first derive the operator from these chosen relations in ConceptNet 5 (e.g., from /r/UsedFor to Op_UsedFor). Then we define the verbalizing function for each operator using the description. For example, the operator Op_UsedFor will leverage the description "A is used for B; the purpose of A is B.", where A and B is replaced with the input start node and tail node.
> > > We will move some other less important content to the appendix to save space to explain these clearly in the Section 3.1.
> > >
> > > 2. Symbolic Executor need neural information of node
> > >
> > > We thank the reviewer to provide a very clear way to describe the process of constructing procedural prompt $P_G$. And yes, it's correct. And yes, the Symbolic Executor does not require the vectorized representation of the node. By "acquiring the **neural** information of each natural language node", we admit it is mis-leading to use the word **neural** here, and it should be more accurate to say "acquiring the **structural** information of each natural language node". The **structural** information describes how the natural language node relates to the neighbor nodes and the task, which is adapted by processing the **neural** information of the node and the task name in previous step (Edge-wise Adaption).
> > > We will update this in our paper to be more clear and re-write the paragraph to more clearly express our implementation as the reviewer describe in the comment.

---

### Official Review · Reviewer_UWNS · 2022-11-01

**Confidence:** 3
**Clarity, Quality, Novelty And Reproducibility:** See strengths and weaknesses.
**Correctness:** 4
**Technical Novelty And Significance:** 2
**Empirical Novelty And Significance:** 3
**Recommendation:** 5

**Strength And Weaknesses:**


The paper is interesting and the topic is very relevant. Planning from LLMs is promising and grounding them in common sense as well as admissible scenes is a core challenge in this area.
The approach of building a large knowledge base and leveraging within an LLM is well founded.
The two large planning datasets are large and good environments to test in. The results show improvements across the board compared to baselines.

The paper however has a few areas for improvement.

(1) The clarity could be improved. Some sections like Section 3 are quite dense and difficult to parse, particularly Section 3.1. Perhaps an earlier overview could help clarify or explicit running examples. Section 3.1 could use it’s own algorithm block and potentially a zoomed in figure of the computation. Figure 3, though a nice overview, is very dense.

(2) Though the performance of PLAN is stronger than baselines, it is a smaller improvement than I would think given the additional complexity and also somewhat difficult to judge how large of an improvement it is. In many of the metrics it seems PLAN outperforms by a few percentage points or in voting it wins 50% of the time. While I acknowledge that this shows that PLAN is performing better, it isn’t clear from these results that it is worth the vast additional complexity compared to baselines. The authors should add an additional metric similar to executability in Huang et al., showing the actual success rate of these plans, as this is the ultimate metric we care about. A few related questions:
* I would be interested to learn more about what the main failure modes are.
* Some results, such as Table 1, seem to outperform baselines with larger models. I’m surprised by this as I would think your approach would be particularly important to add structure when LLM’s are more inaccurate. Do you have any intuition on why this might be?

Minor notes:
* “The Termination Condition is either reaching the max step t or the matching score is below threshold θ. “ Instead of thresholding, one can compare to an end of statement token’s probability.
* Mention in the intro where the common sense external knowledge comes from (though I know it is in Figure 1).
* How do you extract entity names from the task name?
* “show that pre-trained knowledge (D) in LLMs confounds” What is D here?
* “We describe the implementation of such frontdoor adjustment in Section 3.1.” but section 3.1 is 2 pages long, be more specific.
* Table 3 bolds 0.433 in the button right, though GPT3 + Chain outperforms it with 0.471
* “PLAN surpasses powerful baselines (Chain of Thoughts (Wei et al., 2022) and Zero-shot Planner (Huang et al.)) by large margins on both the original and the counterfactual samples.” Are these baselines powerful? They don’t have external information except their prompts.


**Summary Of The Paper:**

This work proposes an approach for procedural planning with LLMs and casual models. The approach first builds a commonsense casual model from an external knowledge base with adjustments. Then, given a query, it builds a task-relevant subgraph, which is provided as a procedural prompt. Finally, this is translated to admissible actions for the LLM to plan. This is demonstrated on two large planning tasks, RobotHow and WIkiHow, and shown to outperform baseline planners.

**Summary Of The Review:**

The paper presents and interesting and novel approach to an impactful area of research. PLAN also shows performance on two large and varied environments. However, the paper is quite dense and could be improved in clarity. The results show PLAN outperforms baselines, but by a small margin given the additional complexity, and the results do not include success rate metrics that may be most crucial for understanding the gains in performance.

---

> ### Author Response · Authors · 2022-11-15
> **Response to Official Review of Paper822 by Reviewer UWNS Part2**
>
> ### Minor Notes
> > **R2M(inor)N(ote)1.**  “The Termination Condition is either reaching the max step t or the matching score is below threshold θ. “ Instead of thresholding, one can compare to an end of statement token’s probability.
>
> We follow the termination condition in Zero-shot Planner (Huang et al.), because the procedural planning is also temporally extended. The end of statement token is also a promising termination condition, we try it for the baselines and our method by appending the final step "End of the task" to all the ground truth procedural plans. However, this termination condition works on the GPT3 base architecture, but not works on the GPT2 and BART. We suppose it's due to that the experiment is conducted under zero-shot setting by prompting the large language models, and thus the end of statement condition is not universally ideal for the large language models.
>
> > **R2MN2.**  Mention in the intro where the common sense external knowledge comes from (though I know it is in Figure 1).
>
> We clarify the external knowledge resources (e.g., ConceptNet) in the introduction of the revision.
>
> > **R2MN3.** How do you extract entity names from the task name?
>
> As mentioned in Section 3.1 (Stage 1): "As for the implementation,  the high-level task name $T$ is Semantically Parsed into the Entity Set $T_E$ with the named entity information in order to extract the representative nodes from the graph. We use NLTK to tokenize and \texttt{pos\_tag} the task text. Then we use the noun (e.g. television), noun phrases (e.g. remote control), and verb phrases (e.g. watch television) as the entity node."
> We re-write the paragraph to highlight the implementation of how we extract entity names from the task name in the revision.
>
> > **R2MN4.** “show that pre-trained knowledge (D) in LLMs confounds” What is D here?
>
> The pre-trained knowledge ($D$) is biased to some correlation existed in the pre-training data, and thus is a confounder. For example, as shown in Figure 2, living room and TV is highly correlated in human-annotated training dataset, $D$ in LLMs is a biased prior that assume $P(TV|*living room*) \rightarrow 1$. However it's not always the case that the next plan of go to living room is watch TV, or that when the task is watch TV, you need to go to living room.
>
> > **R2MN5.** “We describe the implementation of such frontdoor adjustment in Section 3.1.” but section 3.1 is 2 pages long, be more specific.
>
> To achieve our front-door adjustment, we inject external knowledge into LLMs with a neuro-symbolic approach by $3$ stages: 1) Intervention on $T$ and $S_{i-1}$ (Stage1 in Section3.1) 2) Intervention on $P_{i-1}$ (Stage2 in Section 3.1) 3) Full temporal-extended intervention on the aggregation of 1) and 2) (Stage3 in Section 3.1).
> We add above specific description in Section 2.2 in the revision.
>
> > **R2MN6.**  Table 3 bolds 0.433 in the button right, though GPT3 + Chain outperforms it with 0.471
>
> We check the result files again, and this is the full metric output for GPT3 + Chain on WikiHow in Table $3$: {"sentence-bleu": 0.09465079848425888, "wmd": 0.9433102857150133, "bert-score-f": 0.8395531213283539, "rouge-1-f": 0.3936172366157493}. We're sorry that the ROUGE-f1 on WikiHow was occassionally pasted from the nubmers on RobotHow. We've used a script file to automatically check the table nubmers in latex format with the raw text output files to ensure the correctness of all other table numbers. And we fixed the number $0.471$ (this is the ROUGE-f1 on RobotHow) as the correct one $0.393$ (this is the actual ROUGE-f1 on WikiHow) in Table 3 in the revision.
>
> > **R2MN7.** “PLAN surpasses powerful baselines (Chain of Thoughts (Wei et al., 2022) and Zero-shot Planner (Huang et al.)) by large margins on both the original and the counterfactual samples.” Are these baselines powerful? They don’t have external information except their prompts.
>
> The Chain of Thoughts (Wei, et al., 2022 baseline utilize additional annotation of Chain of Thoughts exemplar for each task domain. The Zero-shot Planner (Huang, et al.) baseline utilize additional held-out exemplars for each task domain. And then these two baselines will retrieve the most relevant exemplar to the current task, and use it as the prompt. Thus, these baselines are powerful in a way that they directly use human-written procedural plan exemplar, which already contains procedural knowledge in the task-domain.
> However, we devise a technique that follow the inspiration from our causal analysis of procedural planning to automatically structuralize such knowledge from the external resources and inject into LLMs.
> Surprisingly, our method that do not specify task domain (either WikiHow or RobotHow) still surpass these baselines that leverage the knowledge from the task domain itself.

---

> ### Author Response · Authors · 2022-11-15
> **Response to Official Review of Paper822 by Reviewer UWNS Part1**
>
> ### Weaknesses
>
> > **R2W(eakness)1.** The clarity could be improved. Some sections like Section 3 are quite dense and difficult to parse, particularly Section 3.1. Perhaps an earlier overview could help clarify or explicit running examples. Section 3.1 could use it’s own algorithm block and potentially a zoomed in figure of the computation. Figure 3, though a nice overview, is very dense.
>
> We extend the paragraph at the begging of Section 3 and add an earlier overview paragraph of Section 3.1 with more clarity and explicit running examples in Appedix D.2 to help parse Section 3.
> We expand Algorithm 1 with a fine-grained algorithm block for Secion 3.1 that show the computation flow.
>
> >  **R2W2.** Though the performance of PLAN is stronger than baselines, it is a smaller improvement than I would think given the additional complexity and also somewhat difficult to judge how large of an improvement it is... Important to show the actual success rate of these plans, as this is the ultimate metric we care about.
>
> We thank the reviewer for point out the concerns about the complexity and ultimate evaluation metric above. We'll address them point by point:
>  1. Concern about bringing in complexity for improvement. Whether it's worth the vast additional complexity compared to the baselines.
>
> As can be seen in both automatic and human evaluation, our method consistently surpass the baselines and achieve a very good performance especially in win-tie-lose experiment. As for the complexity, after the thorough causal analysis, we come to a implementation that automatically leverage the external knowledge instead of require additional exemplars.
> And we provide the quantitative analysis of time cost comparison in Table 8 and Table 9 (the last column) in Appendix C.2, indicating the comparable computational complexity of our method and baselines. Thus we claim that the brought in complexity is ignorable and is worth for the improvement we achieve.
>
>  2. Difficult to judge the improvement. An additional metric to show the actual success rate of the plans.
>
> As for the automatic evaluation metric, the most human-correlated one is the BERT-score. But our most important metrics are still human evaluation, especially the Win-Tie-Lose experiment.
> In Table 12, we add a new human metric to show whether the procedural plans can successfully implement the task. The consistent improvement over human metrics that evaluate coverage, order and success rate is more convincing to prove the contribution of our work.
> The details of new evaluation results with new added success rate metric is discussed in a new added paragraph "Evaluation on Success Rate Metric" in Appendix C.2.
>
>
> > A few related questions:
> > **R2W(eakness)2R(elated)Q(uestion)1.** I would be interested to learn more about what the main failure modes are.
>
> We extend Appendix E.1 for deeper discussion about the limitations. Please also refer to the response at **R1Q3.** for the discussion of limitations and what failtures those limitations may result in.
> For example, the predicted procedural plan on task "Turking", which refers to "The act or process of performing small tasks using the Amazon Mechanical Turk service." according to Wiktionary. We compare the predicted procedural plan on this task among baselines and our method: (1) The ground truth plan is "Task: Turking. Step 1: Walk to home office.Step 2: Walk to desk.Step 3: Find chair.Step 4: Sit on chair.Step 5: Find computer.Step 6: Switch on computer" (2) The plan predicted by Chain baseline is empty. (3) The plan predicted by LLMaP baseline is "Task: Turking. Step 1: Put teddybear on oven." (4) Our prediction is "Task: Turking. Step 1: Eat food turkey.  Step 2: Drink water.  Step 3: Sleep." We can see that for the "out-of-knowledge" task, our method also lead failure planning. We assume this is mainly due to the limited knowledge in external resources, as discussed in the Appendix E.1, and this main failure mode can be avoided by introducing larger external resources (e.g, Wiktionary), similar as other knowledge-enriched methods.
> We add above analysis in an additional subsection Appendix E.2 to discuss detailed failture modes and examples with analysis in the revision.
>
> > **R2W2RQ2.** Some results, such as Table 1, seem to outperform baselines with larger models. I’m surprised by this as I would think your approach would be particularly important to add structure when LLM’s are more inaccurate. Do you have any intuition on why this might be?
>
> We suppose GPT3 works better given the structured prompt, since they may see similar data. Thus when the correct structure is provided by our model, the larger models (e.g., GPT3 over GPT2) could benefit more and achieve larger performance gain.

---

### Official Review · Reviewer_W99u · 2022-11-03

**Confidence:** 3
**Correctness:** 3
**Technical Novelty And Significance:** 3
**Empirical Novelty And Significance:** 3
**Recommendation:** 8

**Clarity, Quality, Novelty And Reproducibility:**

My main concern is that the causal analysis is both very thorough but it's also confusing. After trying the causal models in the tool https://causalfusion.net/app, I'm obtaining different estimands. The reason seems to be that the model in Fig 2 is updated after each iteration, fixing the values of the previous variables. That makes it hard to understand some of the statements. For instance, eq (1 and 10) says
P(Pi=p | do(S_{i-1})) = P(Pi=p | do(S_{i-1}))
However, using that tool I obtain something like
P(P_3 ∣ do(S_2))=∑_{P_2​,T} P(P_3​∣S_2,P_2,T)P(P_2,T)
P(P_2 ∣ do(S_1))=∑_{P_2​,T} P(P_3​∣S_2,P_2,T)P(P_2,T)
but if I fix T and P_{i-1}, that is what would happen in greedy decoding with an LLM, then I should obtain P(Pi=p | do(S_{i-1})).

Something similar happens with Eq (9)

However, this makes the analysis hard to follow as the new causal models are not referred in the equations.
There is also a comment on P_{i-1} being copied into P_i that makes things more complicated.

Question:
- Am I right about these concerns?

**Let's keep in mind that the algorithm is just an implementation.**
**In principle, what we want is the prediction of all the do() compounded.**

Perhaps the appendix is the place to clarify this point, explaining:
- what's exactly the new causal graph after each iteration.
- why does it make sense to simplify the graph.
- Clarify which causal graph is related to each equation.

More questions:
- does the causal analysis holds given de "adaption" that was necessary? I'd like to see a more clear causal criticism of that situation.
- Does the use of the front criterion holds given that there could exist other words related to the task?
- Where would this break? The results are just positive but there is not a detailed discussion on limitations.

I think the work is interesting and solid.
My only concern is the one I just mentioned on the clarity of the causal analysis.



**Strength And Weaknesses:**

Strengths
- Principled motivation using causality
- Concrete technical solutions challenges for integrating with the selected source of information
- Ablation study shows the impact is not trivial.
- Idea is novel as far as I know.
	- For instance, this survey appeared after the submission. It does not mention work as specific as this submission: Feder, Amir, Katherine A. Keith, Emaad Manzoor, Reid Pryzant, Dhanya Sridhar, Zach Wood-Doughty, Jacob Eisenstein, et al. “Causal Inference in Natural Language Processing: Estimation, Prediction, Interpretation and Beyond.” Transactions of the Association for Computational Linguistics 10 (October 2022): 1138–58. https://doi.org/10.1162/tacl_a_00511.

Weaknesses
- It's not clear which is the current causal model used at each point. (See question below)
- The paper might be assuming P_i is the only path, but the existence of other paths to be blocked it's not discussed.
- The argument loses a bit in the adaptation plus other decisions on what to retrieve.



**Summary Of The Paper:**

The paper presents a principled algorithm for incorporating symbolic information related to text generation using an LLM. The idea is justified using a causal analysis, so the algorithm is motivated by the front door criteria. The external information is extracted by entity extraction and extracting related information from ConceptNet. The results are concatenated to the prompt and interpreted as conditioning. The human evaluation and metrics indicate the method leads to improvement in three different LLMs.


**Summary Of The Review:**

The paper provides a causality-justified algorithm for retrieving information about a task. Although some points seem to break the causal argument, I suspect this is indeed the explanation for better performance. Some aspects of the causal analysis are hard to follow.

---

> ### Author Response · Authors · 2022-11-15
> **Response to Official Review of Paper822 by Reviewer W99u Part2**
>
> #### Clarity of the algorithm and the do() compounded
> > **R1C3.** What's exactly the new causal graph after each iteration.
>
> Please refer to the "timestep-wise causal model " in the response at **R1W(eakness)1.**, where we explain more on how Figure 2, Figure 4 a show the causal graph after each iteration (timestep-wise). Basically, the causal graph at timestep $i = 1$ is shown in Figure 4(a) and the causal graph at timestep $i > 1$ is shown in Figure 4(b). During each iteration, the system go through several operations, and the operation-wise change on the causal graph is depicted in Figure 5.
>
> > **R1C4.** Why does it make sense to simplify the graph.
>
> We simplify the causal graph at timestep $i$ ($i$ > 1) in Figure 4(b) into Figure 4(c) in Appendix A.3. After simplification based on Equation 12-16, we get the SCM at timestep $i>1$ in Figure 4(b). This is an equivalent SCM after eliminating $P_{i-1}$ in Figure 4(c).
> The reason we could eliminate $P_{i-1}$ is as follows. We follow a common method of constructing a temporally-extended prompt, which is to append the prediction at previous timesteps to the prompt at the current timestep. In our case, the $P_{G, i}$ is the same as $P_{G, i-1}$, thus $P_i$ inherit part of the content from $P_{i-1}$, the change only depend on the $S_{i-1}$. Thus $P_{i-1}$ and $S_{i-2}$ are fixed, and there is no need to predict $P_{i-1}$ at timestep $i$ again. In this way, we simplify the causal graph in Figure 4(b) to the one in Figure 4(c). In summary, we define and simplify the causal graph based on the temporal-extended property of our prompt construction ($P_{i}$ inherits the content from $P_{i-1}$).
> We extend the above clarity of definition paragraph in Appendix A.3.
>
> > **R1C5.** Clarify which causal graph is related to each equation.
>
> Please refer to the "operation-wise causal model" in the response to the response at **R1W(eakness)1.**. We clarify this with Figure 5 in Appendix A.3 about the SCM after each equation (operation-wise).
>
> #### More Questions
> > **R1Q(uestion)1.** does the causal analysis holds given de "adaption" that was necessary? I'd like to see a more clear causal criticism of that situation.
>
> The Edge-wise Adaption is actually an implementation of the $do$-operator on $P_i$ by adjusting the weight of the edge in the external knowledge graph. Please refer the revision and response to **R1W3.** for details of the adaption module. The causal analysis still holds given that the adaption is to intervene $P_i$ as a part of front-door adjustment instead of changing the situation.
>
> > **R1Q2.** Does the use of the front criterion holds given that there could exist other words related to the task?
>
> If I'm understanding your question correctly, you're asking the situation that different words can describe the same task.
> For example, one possible situation is that the words "watch the TV" is the synonym of the words "watch the television".  In this case, with the help of synonym relation recognized in the external resource (e.g., ConceptNet), we could retrieve the same subgraph.
> Another possible situation is that the words related to the tasks are semantically relevant but not recognized as synonymys in the external resources. In this case, since we're still fixing the task $T$ and ground truth procedural plan, the constructed prompt $P_i$ as the mediator intercepts all directed paths from $T$ to $S_i$. Thus the front-door criterion still holds in these situations.
>
> > **R1Q3.**  Where would this break? The results are just positive but there is not a detailed discussion on limitations.
>
> We discuss the limitations in Appendix E.1 from the perspective of the procedural planning task. We extend Appendix E.1 with the limitations in our implementation guided by causal analysis: "First, the coverage of the leveraged external resources is limited, which is common in knowledge-enhanced system. This may result in the wrong understanding of the task and produce not reasonable procedural plans. For example, the knowledge of the word "Turking", which refers to "The act or process of performing small tasks using the Amazon Mechanical Turk service." according to Wiktionary, is not covered in the external resources (e.g., ConceptNet). Since our proposed system do not assume specific external resources. It's doable in the future that we utilize more powerful external resources (e.g., Wiktionary). Second, the hop number and the threshold of the multi-hop retrieval in task-relevant subgraph sampling is currently a configured hyperparameter. This may result in not ideally constructed prompt. The future work could instead make these hyperparameters learnable on each task domain, and also explore the pros and cons between end-to-end commonsense-infused prompt versus neuro-symbolic constructed prompt."
>
> We also add Appendix E.2 to discuss failure modes in the revision.

---

> > ### Comment · Reviewer_W99u · 2022-11-22
> > **Thank you!**
> >
> > Thank you for your detailed answers, and the improvements to the manuscript.
> >
> > Some of the changes are not fully integrated. Just two cases I just noticed:
> > - Fig 5 is not referred to in the text
> > - E.1 refers to E.1
> >
> > I'm still concerned with the statement about the limitations.
> > In general, I don't think the limitations of a method should be in the appendix of a paper.
> > They are almost as important as the contributions.
> >
> > In that sense, R1Q2 and R1Q3 are not telling the whole story.
> > For instance, discussing "watch the television", the authors said: "In this case, with the help of synonym relation recognized in the external resource (e.g., ConceptNet), we could retrieve the same subgraph."
> >
> > I think the papers need to clarify the compounding effects of those situations in the main text.
> > The appendix should include a case where the behaviour is not correct.
> >
> > At this point, the paper should be accepted anyway without my late suggestions.
> > I do think making such changes would be a more solid, serious paper, especially in these times when acknowledging limitations seems to weaken a paper.

---

> > > ### Author Response · Authors · 2022-11-23
> > > **Thank you for your support! We will keep on polishing our paper to be more solid.**
> > >
> > > Thank you for recognizing our work and providing additional suggestions! We would like to address your remaining concerns  to polish our paper as below:
> > >
> > > 1. Minor issues
> > >
> > > We add the reference to Figure 5 in Section 2 and Appendix A.3 and remove the redundant sentence "We extend Appendix E.1 with the above discussions in the revision." in E.1.
> > >
> > > 2. Limitation discussion
> > >
> > > We agree that the limitation discussion is important. In addition to the limitations in the Appendix, we add clear clarification in Section 3 with the limitations of our implementations when illustrating our approach. And we incorporate the response at **R1Q2** in the limitation discussion as well.
> > >
> > > 3. Clarity of compounding effects
> > >
> > > Thank you for pointing this out. And we incorporate responses that involve clarity of compounding effects in Section 2 as suggested.
> > >
> > > Thank you again for supporting our work. And we will sure incorporate all the suggestions to revise our paper to a more solid one!

---

> ### Author Response · Authors · 2022-11-15
> **Response to Official Review of Paper822 by Reviewer W99u Part1**
>
> ### Weaknesses
> > **R1W(eakness)1.** It's not clear which is the current causal model used at each point. (See question below)
>
> As for the timestep-wise causal model (e.g., SCM when predicting Step 1 at each **iteration**), Figure 2(b) show the general SCM at step $i$, and Figure 4(a)(b)(c) show more detail of the difference between the SCM at initial timestep (timestep $i$=1) and the following timesteps (timestep $i$>1). We also add the reference to the SCM for each stage in Algorithm 1 in the revision.
> As for the operation-wise causal model (e.g., SCM before and after each **equation** at each iteration), we add a new Figure 4 in Appendix A.3 to show how the SCM transit before and after each equation.
>
> > **R1W2.**   The paper might be assuming $P_i$ is the only path, but the existence of other paths to be blocked it's not discussed.
>
> We make the assumption of strong ignorability, that there is only one confounder $P_i$. between $T$ and $S_i$. As one assumption of the front-door criterion is that the only way in which the task name $T$ influences $S_i$ is through the mediator $P_i$. Thus $P_i$ must be the only path, otherwise, the front-door adjustment cannot stand. Notice that $D$ already represents all the knowledge in pre-trained data and LLMs. Thus it's reasonable to use the strong ignorability assumption that it already includes all possible confounders.
>
> > **R1W3.** The argument loses a bit in the adaptation plus other decisions on what to retrieve.
>
> The Edge-Wise Adaption is an implementation of the $do$-operator on $P_i$ by adjusting the weight of the edge in the external knowledge graph. The external knowledge graph $G$ already annotates the weight of each edge ($E_w$). The edge link entity nodes in the graph. We argue that the annotated weight will result in a spurious correlation since they capture the task-invariant concept nodes as the causal factors.
> Thus we additionally consider the semantic relatedness between the node embedding in the graph with the task embedding. Specifically, we adapt the original $E_w$ by the addition of $E_w$ and the cosine similarity of node embedding and task embedding as $\hat{E_w} \leftarrow E_w + cosine(n_{E_{tail}}, v_{task})$, where $n_{E_{tail}}$ is the tail node embedding of the edge $R_e$ and $v_{task}$ the task embedding. We re-write the paragraph about Edge-wise Adaption with above details in Stage 2 paragraph in Section 3.1.
> The nodes $N_e$ are retrieved by ranking the adaptative weight $\hat{E_w}$. Since the nodes linked with these highly-weighted edge are most likely to provide commonsense knowledge for generating the procedural plan for the task. We decide to retrieve $3$-hop subgraph for two reasons: 1) there is a limit of input sequence length for the LLMs 2) the downgraded relatedness of the node that is multi-hops away from the entity nodes parsed from the task $T$, as previous work using ConceptNet also observe the same trend and also decide to do $3$-hop subgraph retrieval ([1]).
> We add more details of the adaption and decisions on what to retrieve in Stage 2 paragraph in Section 3.1 and Appendix B.2.
>
> [1] Xikun Zhang, AntoineXikun Zhang, Antoine Bosselut, Michihiro Yasunaga, Hongyu Ren, Percy Liang, Christopher D. Manning, Jure Leskovec, "GreaseLM: Graph REASoning Enhanced Language Models for Question Answering", ICLR 2022
>
> ### Clarity, Quality, Novelty And Reproducibility
> #### Clarity of Causal Analysis
> > **R1C(larity)1.** Clarity of the Equations (Eq.1, Eq.10 (now Eq. 9 in revision) and Eq.9 (now Eq. 8 in revision)) in paper compared with using the causalfusion tool.
>
> We thank the reviewer to bring out the comparison between the estimands from the tool and our equations. And yes, the model in Figure 2 is updated after each iteration with fixed values of previous variables. For example, at timestep $i$, the $T$ and $P_{i-1}$ are fixed and do not require further prediction. In other words, we're actually doing timestep-wise greedy decoding (each step $S_i$ is greedy decoded at each timestep $i$). Notice that at token-wise, we keep the decoding unchanged for LLM.
>
> > **R1C2.** Refer to the causal models in the equations.
>
> We add the details of both timestep-wise causal model (what's the SCM at each timestep) and operation-wise causal model (what's the SCM before and after each equation) in Figure 4 in Appendix A.3. Please also refer to the response at **R1W(eakness)1.**

---

### Author Response · Authors · 2022-11-18
**Summary of Revision**

We thank all the reviewers for their efforts in reviewing our paper. We attach great importance to every suggestion and comment from the reviewers. We address each concern point-wisely and greatly improve the paper in the submitted rebuttal revision.

We summarize the changes in our revision below:
1. We re-write the introduction section, including adding a brief introduction as well as a reference to the Appendix for each technical term (e.g, SCM, mediator, front-door adjustment), framing the motivation of our implementation which is guided by our causal analysis more smoothly, and ask people from different research background to help adjust the confusing part in the paper.
2/ We modify the content in Section 2 with clearer problem definitions and more specific references to the implementation sections.
3. We revise the overview at the beginning of Section 3 and add an additional overview for Section 3.1. And we address the clarity of implementation details in Stage1, Stage2, and Stage3 paragraphs. In addition, we add a zoomed-in computation block for Section 3.1.
4. We add the discussion of interesting related works suggested by reviewers in Section 5.
5. We add more background in Appendix A. Causal Preliminaries.
6. We provide details for the causal graph after $do$-operation in Figure 5 and clarify why it makes sense to simplify the SCM from Figure 4(b) to 4(c) in Appendix A.2 front-door adjustment.
7. We move the decision of choosing conditioned generation over a sequential choice-making approach for procedural planning from the introduction to Appendix B.3 Method Details.
8. We add missing configurations of hop number in Appendix B.4
9. We expand Table 3 with full results for comparison and add more qualitative examples in Table 9-10 and Table 14-16.
10. We add the "Motivation of Evaluation Metrics" paragraph in Appendix C.2.
11. We add an additional evaluation of Success Rate Metrics in Table 11 in Appendix C.2.
12. We add additional ablation study by replacing $\hat{P_{G}}$ with $P_{G}$ in Table 12 in Appendix C.2.
13. We move automatic evaluations on counterfactual task samples to Table 13 in Appendix C.2.
14. We show the intermediate output ($G_s$, $P_G$, $\hat{P_G}$ and $S_T$) of the sample from the evaluation set in Appendix D.1.
15. We expand Appendix E.1 Limitations section with a limitation discussion on our implementation and add Appendix E.2 with failure analysis under possible failure modes.

---

> ### Comment · Reviewer_UGe5 · 2022-11-18
> **Good. We'll take them into account.**
>
> We thank the authors for the response and revision. We will look into that later and discuss the revised version. Thanks.

---

> > ### Author Response · Authors · 2022-11-19
> > **Thank you for your effort on reviewing this paper.**
> >
> > We thank the reviewer for continuing to provide thoughtful suggestions and help us make a better revision. We are glad to address concerns if there are still any.

---

### Comment · Reviewer_UGe5 · 2022-11-28
**I raised my score to a 10.**

I raised my score again because of two reasons.
1. The authors greatly improved their draft, which is also recognized by other reviewers.
2. The negative-side reviewer does not post new comments.

---

### Decision · Program_Chairs · 2023-01-20

**Decision:**

Accept: notable-top-25%

**Justification For Why Not Higher Score:**

Paper could be improved in terms of clarity

**Justification For Why Not Lower Score:**

Impressive results for procedural plan generation using large language models.

**Metareview: Summary, Strengths And Weaknesses:**

The paper presents a new pipeline for symbolic information related to text generation using an LLM which his used for procedural planning. The paper show more detailed and natural generated plans and is evaluated with different large language models.

The paper presents an interesting and principled idea to a very important problem. Moreover, the results are compelling and most reviewers endorse this paper up to the point that they are highly excited. One reviewer is less enthusiastic, but his main concerns (clarity and only small improvement of the performance) have been addressed properly.

**Note From Pc:**

if the above contains the word "oral" or "spotlight" please see: "oral" presentation means -> notable-top-5% and "spotlight" means -> notable-top-25%. As stated in our emails, we are disassociating presentation type from AC recommendations

**Summary Of Ac-Reviewer Meeting:**

NA